# A Survey on Large Language Models for Critical Societal Domains: Finance, Healthcare, and Law

**Zhiyu Zoey Chen**[*], *The University of Texas at Dallas*        *zhiyu.chen2@utdallas.edu*
**Jing Ma**[*], *Case Western Reserve University*        *jing.ma5@case.edu*
**Xinlu Zhang**[*], *University of California, Santa Barbara*        *xinluzhang@ucsb.edu*
**Nan Hao**[*], *Stony Brook University*        *hao.nan@stonybrook.edu*
**An Yan**[*], *University of California, San Diego*        *ayan@ucsd.edu*
**Armineh Nourbakhsh**[*], *Carnegie Mellon University*        *anourbak@andrew.cmu.edu*
**Xianjun Yang**, *University of California, Santa Barbara*        *xianjunyang@ucsb.edu*
**Julian McAuley**, *University of California, San Diego*        *jmcauley@ucsd.edu*
**Linda Petzold**, *University of California, Santa Barbara*        *petzold@ucsb.edu*
**William Yang Wang**, *University of California, Santa Barbara*        *william@cs.ucsb.edu*

**Reviewed on OpenReview:** *https://openreview.net/forum?id=upAWnMgpnH*

## Abstract

In the fast-evolving domain of artificial intelligence, large language models (LLMs) such as GPT-3 and GPT-4 are revolutionizing the landscapes of finance, healthcare, and law: domains characterized by their reliance on professional expertise, challenging data acquisition, high-stakes, and stringent regulatory compliance. This survey offers a detailed exploration of the methodologies, applications, challenges, and forward-looking opportunities of LLMs within these high-stakes sectors. We highlight the instrumental role of LLMs in enhancing diagnostic and treatment methodologies in healthcare, innovating financial analytics, and refining legal interpretation and compliance strategies. Moreover, we critically examine the ethics for LLM applications in these fields, pointing out the existing ethical concerns and the need for transparent, fair, and robust AI systems that respect regulatory norms. By presenting a thorough review of current literature and practical applications, we showcase the transformative impact of LLMs, and outline the imperative for interdisciplinary cooperation, methodological advancements, and ethical vigilance. Through this lens, we aim to spark dialogue and inspire future research dedicated to maximizing the benefits of LLMs while mitigating their risks in these precision-dependent sectors. To facilitate future research on LLMs in these critical societal domains, we also initiate a reading list that tracks the latest advancements under this topic, which will be released and continually updated[1].

**Keywords:** large language models, GPT-4, interdisciplinary research, finance, healthcare, law, ethics

## 1 Introduction

The advent of large language models (LLMs) such as ChatGPT (OpenAI, 2022) and GPT-4 (Achiam et al., 2023) marks a significant milestone in the evolution of artificial intelligence. The research integrating LLMs with various disciplines, i.e., LLM+X, such as math, science, finance, healthcare, law, etc., is starting as a new epoch powered by collaborative endeavors spanning diverse communities. In this survey paper, we offer an exploration of the methodologies, applications, challenges, ethics, and future opportunities of LLMs within **critical societal domains**, including **finance, healthcare**, and **law**. In this paper, we employ the acronym "FHL" to denote these three domains. These domains are major cornerstones of societal function and well-being, each playing a critical role in the fabric of daily life and the broader economic and social

---

[1]https://github.com/czyssrs/LLM_X_papers

systems. They are frequently discussed together due to shared characteristics, including the *reliance on extensive professional expertise*, *highly confidential data*, *extensive multimodal documents*, *high legal risk and strict regulations*, and the *requirement for explainability and fairness.*

**Reliance on Professional Expertise.** These domains require extensive professional knowledge and experience. The finance domain involves complex financial analysis, investment strategies, and economic forecasting, necessitating deep knowledge of financial theories, market behavior, and fiscal policy (Benninga, 2014; Fridson & Alvarez, 2022; Roberts, 1959; Geels, 2013; Franses, 1998; Easterly & Rebelo, 1993). Healthcare requires specialized knowledge in medical sciences, patient care, diagnostics, and treatment planning, and professionals are trained for years in their specific fields (Melnick et al., 2002; Gowda et al., 2014; P. Collins, 1998; Brauer & Ferguson, 2015; Thomas et al., 2022). The legal domain demands a thorough understanding of legal principles, statutes, case law, and judicial procedures, with practitioners spending extensive periods in legal education and training (Hart & Green, 2012; Dworkin, 1986; Sunstein, 2018; Epstein & Sharkey, 2020; Friedman, 2005; Chemerinsky, 2023; MacCormick, 1994).The need for profound professional expertise in these domains presents significant challenges in equipping LLMs with the requisite knowledge and capabilities (Li et al., 2023d; Xie et al., 2024b; Islam et al., 2023; Nori et al., 2023a;b; Tan et al., 2023; Yu et al., 2022; Choi et al., 2021; Iu & Wong, 2023).

**Highly Confidential Data.** Unlike many other domains where data might be more public or less sensitive, FHL domains deal with information that is mostly personal and confidential. This brings unique challenges for LLM-based research, which is essentially data-driven. LLMs must be trained and tested in a manner that prevents data breaches or inadvertent disclosures. This necessitates research challenges such as training data synthesis, encryption techniques, secure data handling practices, transfer learning, etc.

**Extensive Multimodal Documents.** The complexity and multimodal nature of documents in these sectors mark another unique challenge. Financial documents may contain not only text but also tables and charts in diverse structures (Chen et al., 2021b; Bhatia et al., 2024). Healthcare data may contain text and various medical imaging modalities Gee et al. (2004); Wood et al. (2020); Yan et al. (2023c), such as X-ray Radiography, Ultrasound, Computed Tomography (CT) and Magnetic Resonance Imaging (MRI). Legal documents may contain text, images of evidence, audio recordings of testimonies, or video depositions (Matoesian & Gilbert, 2018; He et al., 2023a). Developing LLMs that can accurately interpret and correlate information across modalities is crucial, demanding innovative approaches to model architecture and data processing.

**High Legal Risk and Strict Regulations.** Considering the potentially serious consequences of actions in FHL domains, their regulatory landscape is more complex and stringent than in many other fields. They must adhere to rigorous standards and laws from the outset to protect client welfare and ensure compliance. Such requirements pose unique challenges for developing LLM-based applications, as researchers need to design models carefully to ensure compliance with regulations (Meskó & Topol, 2023; Minssen et al., 2023; Zhang et al., 2023g; Ong et al., 2024; Hacker et al., 2023; Gilbert et al., 2023). LLMs must incorporate mechanisms to ensure regulatory compliance to not only achieve accuracy but also be extraordinarily aware of legal and regulatory nuances.

**Requirement for Explainability and Fairness.** Explainability and fairness have emerged as vital components of AI (Adadi & Berrada, 2018; Burkart & Huber, 2021; Mehrabi et al., 2021; Caton & Haas, 2020; Dong et al., 2023), ensuring transparent decision-making processes and guarding against biased outcomes. Particularly in knowledge-intensive and high-stakes domains like FHL, decision-making often involves professional expertise and complicated processes. Furthermore, these decisions can directly influence people's lives in significant aspects (e.g., economic status, health, and legal rights). These facts necessitate a higher standard of transparency and bias mitigation of model design in these critical societal domains to maintain public trust and compliance with ethical guidelines Beauchamp & Childress (2001); Cranston (1995); Yamane (2020); Svetlova (2022). Developing LLM-based applications that offer transparent reasoning and minimize bias is vital for any real-world deployment in these domains.

Focusing on FHL domains, this paper explores the extensive spectrum of LLM applications, underscoring LLM's transformative effects across these critical societal sectors. This exploration sheds light on how LLMs are reshaping traditional research methodologies in these important fields, and fostering the innovation,

efficiency, and social impact of next-generation AI. More specifically, the rest of paper organization is as follows. We discuss related surveys in Section §2. We investigate the domains of finance, healthcare, and law in Section §3, §4, and §5, respectively. In Section §6, we consider a series of ethical concerns regarding adopting LLMs in these domains. Finally, we make conclusions in Section §7.

## 2 Related Surveys

Along with the rapidly evolving LLM research, there is a surge of LLM-related survey literature that explores a wide range of perspectives and aspects of LLM development. In addition to the surveys investigating the overall development of LLMs (Zhao et al., 2023a; Min et al., 2023a), recent surveys include fine-grained areas such as alignment (Shen et al., 2023a; Wang et al., 2023d; Liu et al., 2023d), augmentation (Mialon et al., 2023; Gao et al., 2023b), instruction tuning (Zhang et al., 2023e), reasoning (Huang & Chang, 2022; Qiao et al., 2022), compression (Zhu et al., 2023a), evaluation (Chang et al., 2023), explainability (Zhao et al., 2024), and hallucination (Zhang et al., 2023l; Huang et al., 2023), as well as bias, fairness, and safety (Gallegos et al., 2023; Navigli et al., 2023; Li et al., 2023e; Weidinger et al., 2021; Yao et al., 2024; Shayegani et al., 2023). Surveys on LLM-based research in NLP tasks are also prevalent, such as text generation (Li et al., 2022a; Zhang et al., 2023c), code generation (Zan et al., 2022), information retrieval (Zhu et al., 2023b), recommendation (Wu et al., 2023c), etc. As the study of LLM+X becomes increasingly popular, surveys in this direction have also begun to emerge in domains such as robotics (Zeng et al., 2023), education (Yan et al., 2024b), software engineering (Fan et al., 2023), causal inference (Liu et al., 2024c), etc. In contrast with existing surveys that mostly focus on LLM integration for NLP tasks or STEM disciplines, our survey investigates LLMs in three critical societal sectors — FHL domains.

In the finance domain, there are existing surveys on AI, machine learning, or deep learning in finance (Cao, 2022; Cao & Zhai, 2022; Maple et al., 2023; Ozbayoglu et al., 2020; Rundo et al., 2019), as well as general NLP techniques in finance (Xing et al., 2018; Fisher et al., 2016; Gao et al., 2021b; Gupta et al., 2020; Kumar & Ravi, 2016). In contrast, our survey focuses on cutting-edge LLM development in finance. The works by Li et al. (2023g) and Lee et al. (2024) address LLM techniques in finance. However, they primarily introduce general LLM techniques, financial-specific LLMs, and financial tasks. In contrast, our survey on finance not only covers more thorough explorations of financial tasks and financial-specific LLMs, but also investigates performance comparisons and analysis for LLMs, offering insights and guidance for future research. Furthermore, our survey explores LLM-based methodologies and adjacent research, concluding with a broad discussion of future prospects, emphasizing their implications for critical societal sectors from a comprehensive range of viewpoints.

In the medical domain, previous studies have extensively explored applications of machine learning (Garg & Mago, 2021; Shehab et al., 2022) and deep learning (Piccialli et al., 2021; Egger et al., 2022; Miotto et al., 2018), with specific emphasis on NLP within medical contexts (Chary et al., 2019; Wu et al., 2020; Liu et al., 2022; Kalyan & Sangeetha, 2020). Our survey broadens the scope by including LLMs and their diverse applications in the medical field. Concurrently, Zhou et al. (2024a); Bedi et al. (2024); Omiye et al. (2024) investigate LLMs in the medical domain focusing primarily on single modality applications, while another work Hartsock & Rasool (2024) focused on medical Visual Question Answering (VQA) and report generation specifically. Our work encompasses a broader spectrum, surveying various applications in both the pure NLP domain and multimodal scenarios. We also discuss recent novel tasks such as medical instruction following and medical imaging classification via natural language.

In the law domain, there are existing surveys on AI in law (Chalkidis & Kampas, 2019; Cui et al., 2023b; Dias et al., 2022), our work improves the focus on current developments in LLMs within the legal area. While the studies by Katz et al. (2023) and Sun (2023) provide an overview of LLM techniques in legal contexts, they primarily discuss generalized LLM techniques alongside legal-specific LLMs and tasks. Our analysis extends beyond these initial explorations, offering a comprehensive examination of legal tasks catered to by legal-specific LLMs and conducting in-depth performance comparisons and analytical reviews of LLMs. This affords pivotal insights and directional guidance for burgeoning research. Furthermore, our survey explores LLM-based methodologies and allied areas of study, ultimately leading to an expansive discourse on future prospects. We place particular emphasis on the augmentation of datasets and the consideration of

non-structured knowledge. Additionally, we underscore the importance of enhancing LLM interpretability and the integration of ancillary tools, which together forecast a revolutionary impact on key societal legal institutions.

With regard to ethics, there have been a few systematic surveys (Khan et al., 2022; Kaur et al., 2022; Mehrabi et al., 2021; Caton & Haas, 2020; Dong et al., 2023) and specifically ethics in LLMs (Liu et al., 2023d; Sun et al., 2024; Ray, 2023; Yao et al., 2024). In contrast to most of these surveys, which introduce ethics or trustworthiness in a general sense, in this work, we focus on ethics in the critical FHL domains. More specifically, we highlight several ethical principles and considerations that are considered most important in these domains, showcase unique definitions and examples of ethics from different domains under these ethical principles, and summarize the progress of existing domain-specific studies for LLM ethics in the three domains respectively.

**Differences from existing surveys**. We summarize the main differences between our survey and existing ones as follows:

- **Scope**. Unlike existing surveys that predominantly explore LLMs in general areas, our study uniquely focuses on LLMs across the three critical societal sectors of FHL. More specifically, our study not only offers a unified high-level overview of the common ground of FHL areas, but also provides an in-depth review within each sector. This dual perspective ensures that our paper stands apart from general LLM-related surveys, as well as studies focusing on single application domains.

- **Depth**. Our survey delves deeper than existing literature of LLMs in within FHL domains. Each sector includes a thorough review covering tasks, techniques, evaluations, future prospects, and domain-specific ethics. The depth and breadth are both more extensive and integrative than existing related surveys.

- **Contribution**. (1) To the best of our knowledge, we are the first to provide a comprehensive view of LLM across the FHL sectors and highlighting their importance, connections, and challenges. (2) Our work meticulously reviews and organizes existing research into a well-structured categorization that spans problems, methodologies, experiments, critical analyses, ethical discussions specific to each sector. (3) We identify and outline promising future research directions in LLM studies within the FHL domains.

## 3 Finance

In this Section, we introduce the existing NLP tasks in the finance domain, including task formulations and datasets. In §3.2, we investigate various Pre-trained Language Models (PLMs) and LLMs developed for finance. In §3.3, we study the evaluations and analysis of the performance of various LLMs. In §3.4, we study various LLM-based methodologies developed for financial tasks and challenges. Finally, we summarize insights, make conclusions, and discuss potential future directions.

### 3.1 Tasks and Datasets in Financial NLP

In this section we introduce existing financial tasks and datasets studied extensively using LLM-related methods, including sentiment analysis, information extraction, question answering, text-enhanced stock movement prediction, and others. We also discuss additional financial NLP tasks that are mostly under-explored for LLM-based methods, suggesting future research opportunities. Figure 1 provides a summary of existing financial NLP tasks.

**Sentiment Analysis (SA).** The task of financial sentiment analysis aims at analyzing textual data related to finance, such as news articles, analyst reports, and social media posts, to gauge the sentiment or mood conveyed about specific financial instruments, markets, or the economy as a whole. An automatic analysis of the sentiments can help investors, analysts, and financial institutions to make more informed decisions by providing insights into market sentiment that might not be immediately apparent from quantitative data.

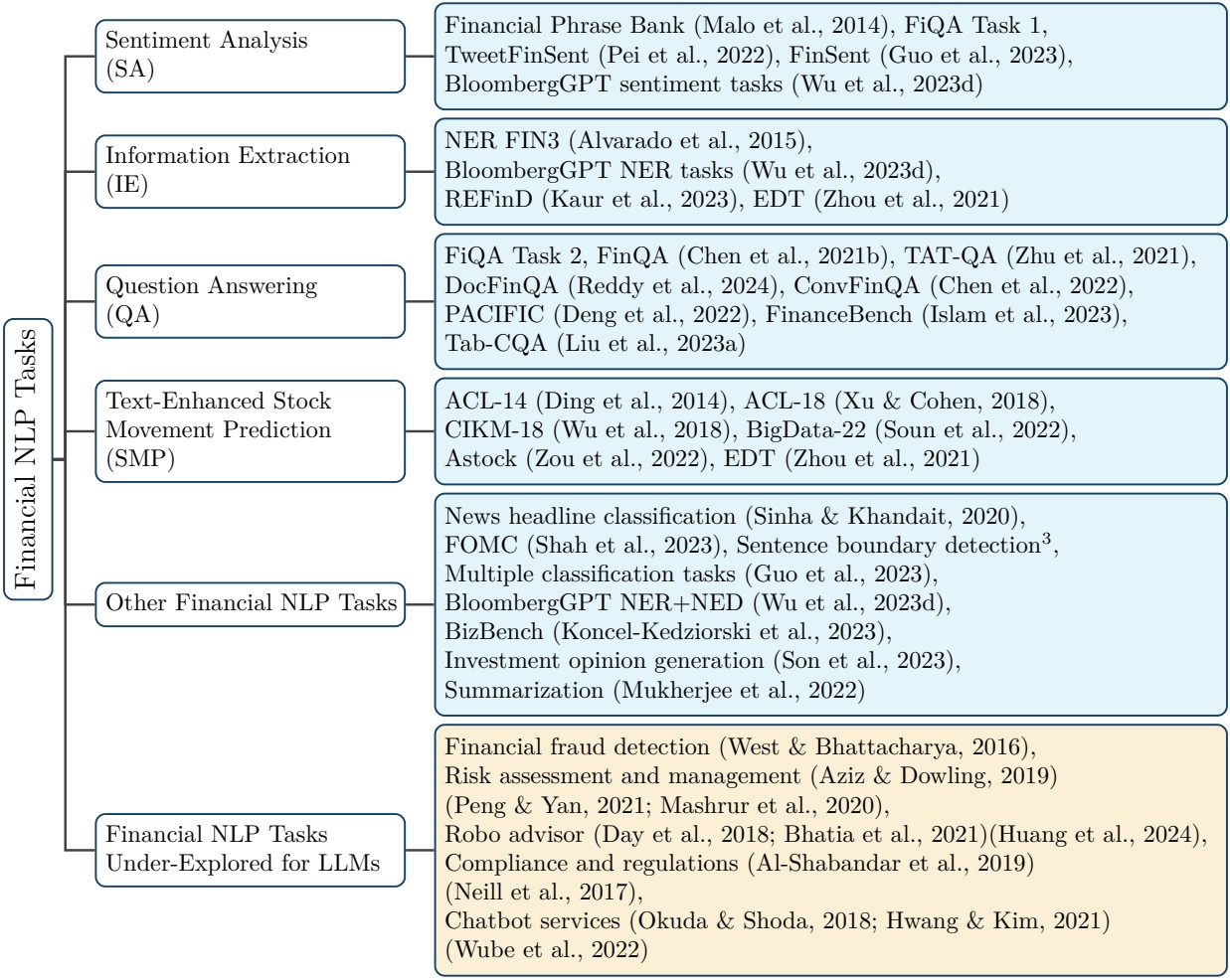

Figure 1: A summarization of existing financial NLP tasks and representative datasets. The yellow field shows the tasks relatively under-explored for LLMs.

The task of financial sentiment analysis is often formulated as a classification problem, with the input as the text to be analyzed and the target label as sentiment orientations such as positive, negative, or neutral. The Financial Phrase Bank dataset (Malo et al., 2014) is based on company news (in English), with the target sentiment categories from the investor's perspective. The FiQA[2] Task 1 focuses on aspect-based financial sentiment analysis, where the target is given as continuous numeric values. TweetFinSent (Pei et al., 2022) is another dataset based on stock tweets. The authors propose a new concept of sentiment labels indicating the opinion of stock movement forecasting. FinSent (Guo et al., 2023) is another sentiment classification dataset based on sentences from analyst reports of S&P 500 firms. In the evaluation of the financial language model BloombergGPT (Wu et al., 2023d), the authors also propose a set of sentiment analysis datasets.

**Information Extraction (IE).** Information extraction involves several key tasks that are essential for analyzing and understanding financial texts. **Named Entity Recognition (NER)** targets identification and classification of key entities in text, such as company names, stock symbols, financial metrics, and monetary values. In (Alvarado et al., 2015), a NER dataset is proposed, aiming at extracting fields of interest for risk assessment in financial agreements. In BloombergGPT (Wu et al., 2023d), the internal financial datasets proposed include NER over various sources. **Relation Extraction (RE)** focuses on identifying and categorizing finance-specific semantic relationships between the entities, such as *cost_of*,

---

[2]https://sites.google.com/view/fiqa/home/

[3]https://sites.google.com/nlg.csie.ntu.edu.tw/finnlp/shared-task-finsbd

*acquired_by.* REFinD (Kaur et al., 2023) is a large-scale RE dataset built upon the 10-X filings from the Securities and Exchange Commission (SEC), focusing on common finance-specific entities and relations. **Event Detection** involves identification of significant financial occurrences like acquisitions, earnings reports, or stock repurchases, from sources such as news or social media. The EDT (Zhou et al., 2021) dataset focuses on detecting corporate events from news articles, aiming at predicting stock movements. In (Oksanen et al., 2022), the authors propose building knowledge graphs over SEC filings. These information extraction tasks play a fundamental role in transforming raw financial texts into structured, actionable insights, aiding professionals in more effective and efficient analysis.

**Question Answering (QA).** Question answering (QA) in finance involves building systems to answer finance-specific queries, typically from large volumes of financial data such as online forums, blogs, news, etc. Such QA systems can aid professionals in performing efficient financial analysis and decision-making. The FiQA[4] Task 2 is an early financial QA dataset targeting opinion-based QA over microblogs, reports, and news. Due to the fact that company financial documents contain large amounts of numeric values, which are of great importance for analysis and decision-making, later works have begun to explore complex QA involving numerical reasoning. The FinQA (Chen et al., 2021b) dataset offers expert annotated QA pairs over earnings reports of S&P 500 companies. The questions require complex numerical reasoning over both the textual and tabular contents in the reports to answer. TAT-QA (Zhu et al., 2021) is another large-scale QA dataset over financial reports containing both textual and tabular contents. In addition to numerical reasoning questions with arithmetic expressions to answer, the dataset also contains extractive questions whose ground truth answers are span or multiple spans from the input report. DocFinQA (Reddy et al., 2024) extends FinQA to the long-context setting, aiming to explore long-document financial QA. ConvFinQA (Chen et al., 2022) extends FinQA to a conversational QA setting involving numerical reasoning over the entire conversation history to capture long-term dependencies. PACIFIC (Deng et al., 2022) is another conversational QA dataset built upon TAT-QA (Zhu et al., 2021), focusing on building proactive assistants that ask clarification questions and resolve co-references. FinanceBench (Islam et al., 2023) is a large-scale QA dataset focusing on the open-book setting covering diverse sources and scenarios. Tab-CQA (Liu et al., 2023a) is a tabular conversational QA dataset created from Chinese financial reports of listed companies in a wide range of sectors.

**Text-Enhanced Stock Movement Prediction (SMP).** Text-enhanced stock movement prediction involves analyzing financial texts like news, reports, and social media to forecast stock price trends and market behavior. The task is mostly formulated to predict stock movement for a target day based on the financial texts and historical stock prices in a time window. In (Ding et al., 2014), the authors propose using extracted events to make stock predictions, and they construct a dataset for Standard & Poor's 500 stock (S&P 500) based on financial news from Reuters and Bloomberg. In (Xu & Cohen, 2018; Soun et al., 2022; Wu et al., 2018), datasets are proposed based on stock-specific tweets from Twitter. Astock (Zou et al., 2022) is a Chinese dataset providing stock factors for each stock, such as Price to Sales ratio, turnover rate, etc. In the EDT (Zhou et al., 2021) dataset, the authors propose to make stock predictions immediately after a news article is published and to perform tradings. The proposed EDT dataset includes minute-level timestamps and detailed stock price labels.

**Other Financial NLP Tasks.** In (Sinha & Khandait, 2020), the authors propose a dataset for classification of news headlines regarding the gold commodity price into semantic categories such as price up and price down. In (Shah et al., 2023), the authors propose a dataset on hawkish-dovish classification based on monetary policy pronouncements by the Federal Open Market Committee (FOMC). The FinSBD-2019 Shared Task[5] proposes the task of financial sentence boundary detection, with the goal of extracting well-segmented sentences from financial prospectuses. In (Guo et al., 2023), the authors propose an evaluation framework including a set of proprietary classification datasets. In BloombergGPT (Wu et al., 2023d), one of the internal evaluation tasks proposed is NER+NED, namely NER followed by named entity disambiguation (NED). The goal is first to identify company mentions in financial documents and then to generate the corresponding stock ticker. In the recent BizBench (Koncel-Kedziorski et al., 2023) benchmark, the authors aim to evaluate financial reasoning abilities and propose eight quantitative reasoning datasets. In (Son et al., 2023), the authors propose the

---

[4]https://sites.google.com/view/fiqa/home/
[5]https://sites.google.com/nlg.csie.ntu.edu.tw/finnlp/shared-task-finsbd

| Model Name | Model Architecture | Evaluation Tasks | Languages | Size | Year |
|---|---|---|---|---|---|
| Pre-training & Downstream task fine-tuning approaches | | | | | |
| FinBERT-19 (Araci, 2019) | BERT | SA | English | 110M | 2019 |
| FinBERT-20 (Yang et al., 2020) | BERT | SA | English | 110M | 2020 |
| FinBERT-21 (Liu et al., 2021) | BERT | SA, QA, Others | English | 110, 340M | 2021 |
| Mengzi-BERTbase-fin (Zhang et al., 2021) | RoBERTa | IE, Others | Chinese | 103M | 2021 |
| FLANG (Shah et al., 2022) | BERT, ELECTRA | SA, IE, QA, Others | English | 110M | 2022 |
| BBT-Fin (Lu et al., 2023) | T5 | SA, IE, QA, Others | Chinese | 220M, 1B | 2023 |
| Pre-training approaches | | | | | |
| BloombergGPT (Wu et al., 2023d) | BLOOM | SA, IE, QA, Others | English | 50B | 2023 |
| Instruction fine-tuning approaches | | | | | |
| FinMA (Xie et al., 2023) | LlaMA | SA, IE, QA, SMP, Others | English | 7B, 30B | 2023 |
| Instruct-FinGPT (Zhang et al., 2023a) | LlaMA-7B | SA | English | 7B | 2023 |
| InvestLM (Yang et al., 2023a) | LlaMA-65B | SA, QA, Others | English | 65B | 2023 |
| FinGPT (Wang et al., 2023c) | Six 7B models | SA, IE, Others | English | 6B, 7B | 2023 |
| CFGPT (Li et al., 2023c) | InternLM-7B | Others | Chinese | 7B | 2023 |
| DISC-FinLLM (Chen et al., 2023b) | Baichuan-13B | SA, IE, QA, Others | Chinese | 13B | 2023 |
| FinMA-ES (Zhang et al., 2024b) | LlaMA2-7B | SA, IE, QA, SMP, Others | English, Spanish | 7B | 2024 |
| FinTral (Bhatia et al., 2024) | Mistral-7B | SA, IE, QA, SMP, Others | English | 7B | 2024 |

Table 1: Summary of financial pre-trained language models. For evaluation tasks, we have **SA** for sentiment analysis, **IE** for information extraction, **QA** for question answering, **SMP** for text-enhanced stock movement prediction, and **Others** for other tasks out of the above three major categories. For the time of release, we report the initial release year of each work.

task of financial investment opinion generation based on analyst reports to evaluate the LLMs' ability to conduct financial reasoning for investment decision-making. In (Mukherjee et al., 2022), the authors propose a dataset for bullet-point summarization from long earnings call transcripts.

**Financial NLP Tasks Under-Explored for LLMs.** The above four categories summarize the current tasks and datasets covered in LLM-related studies. The overall financial NLP space is broader and still has many existing tasks under explored for LLMs. Financial fraud detection is a critical issue with severe consequences in financial activities. There are numerous studies spanning data mining and NLP techniques for financial fraud detection (West & Bhattacharya, 2016; Throckmorton et al., 2015; Seemakurthi et al., 2015; Goel & Uzuner, 2016; Pandey, 2017; Chen et al., 2017; Boulieris et al., 2023; Craja et al., 2020; Calafato et al., 2016), such as detecting fraud in transactions, financial statements, annual reports, tax, etc. Research studies on financial fraud detection using LLM-based methods are largely under-explored. Other tasks remaining relatively open for LLM research include financial risk assessment and management (Aziz & Dowling, 2019; Peng & Yan, 2021; Mashrur et al., 2020; Cheng et al., 2021; Li et al., 2020a; Zou et al., 2017; Giudici, 2018), robo advisor (Day et al., 2018; Bhatia et al., 2021; Huang et al., 2024), compliance and regulations (Al-Shabandar et al., 2019; Neill et al., 2017), chatbot services (Okuda & Shoda, 2018; Hwang & Kim, 2021; Wube et al., 2022), etc. These tasks mostly lack well-defined formulations and well-established public datasets due to their complexity. Given their importance in the financial sector, they are all valuable future directions for incorporating LLM-based research.

## 3.2 Financial LLMs

Since the invention of BERT (Devlin et al., 2019), there have been numerous efforts to build PLMs and LLMs specialized for finance. These LMs typically utilize the same modeling architecture as those designed for general domains. Most works conduct continuous training over existing pre-trained models on the general domain; a few works pre-train the model from scratch using the financial corpus like BloombergGPT (Wu et al., 2023d). The primary distinctions for different financial LMs arise from the training data and the specific training paradigms employed. Consistent with the evolving training paradigms of general PLMs and LLMs, early financial PLMs adopted the pre-training followed by downstream task fine-tuning paradigm and trained relatively small language models. Recent works scaled the model sizes up and conducted instruction fine-tuning, with the evaluation covering broader sets of financial tasks. Most existing financial LLMs are

in single text modality, either in English or Chinese. Table 1 summarizes the PLMs and LLMs for the financial domain, categories by three typical training paradigms: pre-training & downstream task fine-tuning, pre-training, and instruction fine-tuning.

**Pre-Training and Downstream Task Fine-Tuning PLMs.** FinBERT-19 (Araci, 2019) was an early attempt to build financial pre-trained language models targeting the task of financial sentiment analysis. The authors first conducted further pre-training on BERT (Devlin et al., 2019) using a financial corpus, followed by fine-tuning using task training data. FinBERT-20 (Yang et al., 2020) is another PLM on sentiment analysis using similar training strategies, including further pre-training on BERT and pre-training from scratch. FinBERT-21 (Liu et al., 2021) is pre-trained on a large scale of both general domain and financial domain corpus from scratch based on BERT architecture, with a set of self-supervised pre-training tasks. Mengzi-BERTbase-fin (Zhang et al., 2021) is a Chinese model based on the RoBERTa (Liu et al., 2019) architecture, pre-trained with general web corpus and financial domain corpus. FLANG (Shah et al., 2022) is another English model based on BERT (Devlin et al., 2019) and ELECTRA (Clark et al., 2020) using pre-training techniques in ELECTRA. BBT-Fin (Lu et al., 2023) is a Chinese pre-trained language model based on the T5 (Raffel et al., 2020) architecture and pre-training schema. The authors propose a large pre-training financial corpus in Chinese as well as a set of Chinese financial benchmarks, including classification and generation tasks.

**Pre-training LLMs.** BloombergGPT (Wu et al., 2023d) is a large English financial language model built by Bloomberg. As rich financial resources can be obtained within Bloomberg, there is a vast amount of well-curated company financial documents employed for model pre-training, including web content, news articles, company filings, press releases, Bloomberg-authored news, and other documents such as opinions and analyses. Together with three public general domain datasets - The Pile (Gao et al., 2021a), C4 (Raffel et al., 2020) and Wikipedia, the authors created a large training corpus with over 700 billion tokens. The model architecture is based on BLOOM (Scao et al., 2022). The authors note that the common approach of greedy sub-word tokenization (Sennrich et al., 2016; Wu et al., 2016) is not efficient for financial tasks, as it does not handle numeric expressions very well. Instead, they use a unigram-based approach inspired by (Kudo, 2018). In addition, as a pre-processing step, they separate numeric and alphabetic unigrams. To assess the model performance, the authors conducted evaluations based on both external tasks with public datasets and internal tasks with datasets annotated by Bloomberg financial experts. All models are evaluated using vanilla few-shot prompting. For external tasks, including sentiment analysis, headline classification, NER, and conversational QA, BloombergGPT achieved significant improvements for all tasks except NER, over baseline LLMs of similar size. For internal tasks, including sentiment analysis, NER, and NER+NED, BloombergGPT outperforms baseline LLMs for most datasets except NER. For NER, BloombergGPT slightly underperforms the larger 176B BLOOM (Scao et al., 2022) model. Notably, despite increasing the vocabulary size, BloombergGPT's tokenization method improves the efficiency of token representations, and the model outperforms open-domain LLMs on financial QA tasks that involve numerical reasoning. Due to data leakage concerns, BloombergGPT has not been released for public usage.

**Instruction Fine-Tuning LLMs.** FinMA (Xie et al., 2023) is an open-sourced financial LLM built from instruction fine-tuning on LlaMA (Touvron et al., 2023a). The authors note that financial data is often expressed in multimodal contexts such as tables and time-series representations. They develop FLARE, a large instruction-tuning dataset that covers 136 thousand examples from a collection of diverse financial tasks, and contains instructions over tabular and time-series data. For evaluation, FinMA-30B outperforms BloombergGPT and GPT-4 in classification tasks like sentiment analysis and headline classification. Despite being tuned on FLARE, FinMA falls short of GPT-4 (Achiam et al., 2023) on quantitative reasoning benchmarks such as FinQA (Chen et al., 2021b) and ConvFinQA (Chen et al., 2022). The reason could be the lack of proper numerical reasoning process generation data in the instruction-tuning dataset. Instruct-FinGPT (Zhang et al., 2023a) is a model built over LlaMA-7B (Touvron et al., 2023a) using instruction fine-tuning, specifically targeting the task of financial sentiment analysis. CFGPT (Li et al., 2023c) is a Chinese model based on InternLM with continued pre-training and instruction fine-tuning. Following the Superficial Alignment Hypothesis (Zhou et al., 2023), InvestLM (Yang et al., 2023a) is trained with instruction fine-tuning using a well-curated set of 1.3k examples over LlaMA-65B. The resulting model achieves comparable performance and sometimes surpasses the proprietary models like GPT-3.5. DISC-FinLLM (Chen

| Model | FPB | | FiQA-SA | Headline | NER FIN3 |
|---|---|---|---|---|---|
| | Accuracy | F-1 | Weighted F-1 | Weighted F-1 | Entity F-1 |
| Fine-tuning | 0.86 | 0.84 | 0.87 | 0.95 | 0.83 |
| BloombergGPT (Wu et al., 2023d) (few-shot) | - | 0.51 | 0.75 | 0.82 | 0.61 |
| LlaMA-65B (zero-shot) | - | 0.38 | 0.75 | - | - |
| InvestLM (Yang et al., 2023a) (zero-shot) | - | 0.71 | 0.90 | - | - |
| FinMA-30B (Xie et al., 2023) (zero-shot) | 0.87 | 0.88 | 0.87 | 0.97 | 0.62 |
| GPT-3.5 (zero-shot) | 0.78 | 0.78 | 0.76 | 0.72 | 0.29 |
| GPT-3.5 (few-shot) | 0.79 | 0.79 | 0.78 | 0.75 | 0.52 |
| GPT-4 (zero-shot) | 0.83 | 0.83 | 0.87 | 0.84 | 0.36 |
| GPT-4 (few-shot) | 0.86 | 0.86 | 0.88 | 0.86 | 0.57 |

Table 2: Performance comparisons for sentiment analysis tasks (FPB dataset (Malo et al., 2014) and FiQA-SA dataset[8]), headline classification task (Headline dataset (Sinha & Khandait, 2020)), and IE task (NER FIN3 dataset (Alvarado et al., 2015)). The few-shot setting is five shots for FPB, FiQA-SA, and Headline, and twenty shots for NER FIN3. For *Fine-tuning* the model, we select the best performance achieved through fine-tuning models in each dataset respectively. We aggregate the results from (Li et al., 2023d; Xie et al., 2023; Yang et al., 2023a). Note that some results reported for the same model in the above three papers differ, mostly for GPT-3.5 and GPT-4, which may need further verifications.

et al., 2023b) is a Chinese model based on instruction fine-tuning on Baichuan-13B[6], that performs instruction fine-tuning on separate LoRA (Hu et al., 2022) modules for each type of task. FinGPT (Wang et al., 2023c) is a series of 6B/7B models trained with instruction fine-tuning. In (Liu et al., 2023c), the authors propose a framework FinGPT consisting of data collection and processing pipeline from diverse sources, as well as financial LLM fine-tuning using reinforcement learning with stock prices. Most recently, researchers have begun to explore financial LLMs in broader settings. In (Zhang et al., 2024b), the authors propose instruction datasets, fine-tuned model FinMA-ES, and evaluation benchmarks in a bilingual setting of Spanish and English. FinTral (Bhatia et al., 2024) is a series of multimodal LLMs based on Mistral-7B (Jiang et al., 2023). The authors incorporate tool usage (Schick et al., 2023), retrieval-augmented generation (RAG) (Lewis et al., 2020), and visual understanding based on CLIP (Radford et al., 2021a). This allows the authors to explore multimodal contexts, and to include visual reasoning tasks such as question answering over charts and graphs. Despite being pre-trained and instruction-tuned on multimodal data, the FinTral-DPO model underperforms on visual reasoning tasks compared to other SotA multimodal LLMs such as Qwen-VL-Plus (Bai et al., 2023), and GPT-4V (Achiam et al., 2023), but is on par or better than open-source LLMs of similar size. These challenges point to the need for cross-disciplinary research at the intersection of Multimodal LLMs (MLLMs), quantitative reasoners, and financial LLMs.

### 3.3 Evaluation and Analysis

**Performance Evaluation and Analysis for Popular Financial Tasks.** In (Li et al., 2023d; Xie et al., 2023; Yang et al., 2023a), the authors conducted experiments on several popular financial datasets using an array of methods. In this survey, we summarize the performances of representative LLMs on five widely recognized datasets, which have been validated by the research community for their high quality and are frequently employed as benchmarks. Table 2 summarizes the performance of various methods on two sentiment analysis datasets, one headline classification task, and one NER dataset. For sentiment analysis, GPT-4 and the recent instruction fine-tuning models like FinMA (Xie et al., 2023) achieve similar performance as the best fine-tuning methods. For headline classification, FinMA also slightly surpasses the best fine-tuning method. We anticipate that the performances on such datasets have nearly reached saturation. As suggested by (Li et al., 2023d), adopting generalist LLMs could be an easy choice for relatively simple financial tasks.

---

[6]https://github.com/baichuan-inc/Baichuan-13B
[8]https://sites.google.com/view/fiqa/home/

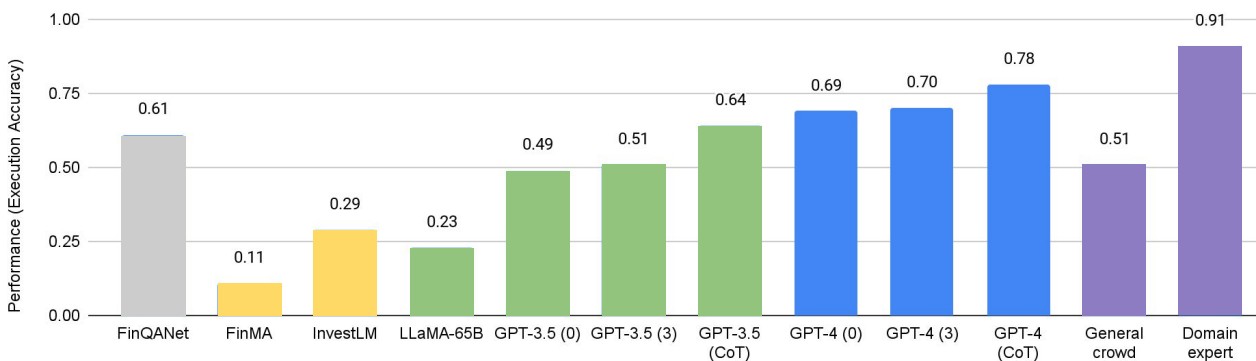

Figure 2: Performance comparison on the FinQA dataset (Chen et al., 2021b). We compare the execution accuracy following the evaluation standard in the original paper. The fine-tuning method Fin-QANet is the RoBERTa-based model in (Chen et al., 2021b); The instruction fine-tuning methods include FinMA (Xie et al., 2023) and InvestLM (Yang et al., 2023a); The general-purpose LLMs include LlaMA-65B, GPT-3.5 and GPT-4, with zero-shot (0), few-shot (3 shots), and CoT prompting; We also list the human expert and general crowd performances. Results are sourced from (Li et al., 2023d; Xie et al., 2023; Yang et al., 2023a).

For IE tasks like NER, there is still a gap between LLMs and fine-tuning methods. On the relation extraction dataset REFinD (Kaur et al., 2023), CPT-3.5 and GPT-4 still largely fall behind the fine-tuning model (Li et al., 2023d).

Figure 2 shows the performance comparisons between various methods on the FinQA dataset (Chen et al., 2021b). Large, general LLMs like GPT-4 still achieve the leading performances with simple prompts, due to the strong knowledge and reasoning ability achieved during pre-training. Domain-specific fine-tuning models follow behind. Instruction fine-tuning models fall far behind compared to the former. Note that in the construction of instruction fine-tuning data in the FinMA (Xie et al., 2023) model, the authors did not include the generation of reasoning programs in the FinQA dataset (Chen et al., 2021b), which could be a major reason for the inferior performance on the FinQA dataset. Therefore, we anticipate that there is still ample room for developing open-source instruction fine-tuning models to improve on tasks requiring complex reasoning abilities.

As demonstrated by (Li et al., 2023d), in most financial tasks, GPT-4 can achieve an over 10% performance increase over ChatGPT. Except for the QA task on FinQA (Chen et al., 2021b) and ConvFinQA (Chen et al., 2022), ChatGPT and GPT-4 perform either comparable or less effective than task-specific fine-tuned models. For FinQA (Chen et al., 2021b) and ConvFinQA (Chen et al., 2022), the authors argue that the reasoning complexity involved is still deemed as basic in financial analysis, but ChatGPT and GPT-4 still make simple errors. Significant improvement is required to adopt these LLMs as trustworthy financial analyst agents in real-world industry usages.

**New Evaluation Frameworks and Tasks.** In (Guo et al., 2023), a financial language model evaluation framework, FinLMEval, is proposed consisting of a set of classification tasks and NER. The authors compare the performances of fine-tuned encoder-only models, such as BERT (Devlin et al., 2019) and RoBERTa (Liu et al., 2019), and zero-shot decoder-only models, such as GPT-4 (Achiam et al., 2023) and FinMA (Xie et al., 2023). Though achieving considerable performance, the zero-shot decoder-only models mostly fall behind fine-tuned encoder-only models on these tasks. The performance gap between the fine-tuned encoder-only models and zero-shot decoder-only models is larger on their proposed proprietary datasets than on public datasets. The authors conclude that there remains room for enhancement for more advanced LLMs in the financial NLP field. Most recently, in (Xie et al., 2024b), the authors propose a large collection of evaluation benchmarks for financial tasks containing 35 datasets across 23 financial tasks. They conclude that GPT-4

mostly performs the best in quantification, extraction, understanding, and trading tasks, and the recent Google Gemini (Team et al., 2023) leads in generation and forecasting tasks.

In (Son et al., 2023), the authors propose the task of financial investment opinion generation based on analyst reports, to evaluate the ability of various LLMs, with and without instruction fine-tuning, to conduct financial reasoning for investment decision-making. The authors conducted experiments on a series of 2.8B to 13B models and concluded that the ability to generate coherent investment opinion first emerges at 6B models, and obtains improvements with instruction-tuning or larger datasets. In (Lopez-Lira & Tang, 2023), the authors study the ability of LLMs to predict stock market returns. It is found that GPT-4 outperforms other LLMs being studied on forecasting returns and delivers the highest Sharpe ratio, suggesting the great potential of advanced LLMs in the investment decision-making process. In (Zhou et al., 2024b), the authors proposed the Financial Bias Indicators (FBI) framework to evaluate the financial rationality of LLMs, including belief bias and risk preference bias. They find that model rationality increases with model size and is often influenced by the temporal bias in the financial training data. Prompting methods, such as instructional and Chain-of-Thought (CoT) (Wei et al., 2022), can also mitigate the biases. In (Islam et al., 2023), the authors propose the task of open-book QA to test the model's ability to handle long context. They conclude that current strong LLMs like GPT-4-Turbo still fall far behind satisfactory performances, either with a retrieval system or using a long context model. In (Callanan et al., 2023), the authors evaluate the ability of GPT-3.5 and GPT-4 in passing the first two levels of the Certified Financial Analyst (CFA) exam. Expectedly, GPT-4 outperforms GPT-3.5 in both levels, but both models struggle with longer contexts, sophisticated numerical reasoning, and tabular information, especially in Level II. The authors also demonstrate that CoT prompting offers limited improvement over zero-shot settings, but in-context learning with 2 or more examples produces the best results. A detailed error analysis reveals that a lack of domain knowledge leads to the majority of errors for both models, especially in Level II exams.

### 3.4 LLM-based Methodologies for Financial Tasks and Challenges

This section addresses LLM-based methodologies that have been proposed to tackle some of the key challenges in Financial NLP, including the scarcity of high-quality data in the public domain, the multimodal nature of many financial documents, the challenge of quantitative reasoning, the lack of domain knowledge in LLMs, and the importance of detecting or preventing hallucinations.

**Confidentiality and Scarcity of High-Quality Data.** Due to the confidential nature of data in the financial domain, clean and high quality datasets can be difficult to obtain (Assefa et al., 2020; Zhang et al., 2023b). In (Aguda et al., 2024), the authors assess the efficacy of LLMs in annotating data for a financial relation extraction task. While larger LLMs such as GPT-4 (Papailiopoulos, 2023) and PaLM-2 (Anil et al., 2023) outperform crowdsourced annotations, they fall far behind expert annotators, demonstrating that domain knowledge plays a crucial role. Other studies have tackled the scarcity of non-English financial training datasets (Zhang et al., 2024b; Hu et al., 2024).

**Quantitative Reasoning.** Reasoning over numerical data is a major component of QA and IE tasks in the financial domain. Several recent studies have proposed prompting strategies that enhance the quantitative reasoning capabilities of LLMs in financial QA tasks.

In (Wang et al., 2024b), the authors introduce ENCORE, a method that decomposes the numerical reasoning steps into individual operations, and grounds each operand within the input context. When used as a few-shot prompting strategy, ENCORE improves the performance of SotA LLMs on TAT-QA (Zhu et al., 2021) and FinQA (Chen et al., 2021b) by an average of 10.9% compared to standard CoT prompting (Wei et al., 2022). In (Chen et al., 2023c), the authors propose the Program-of-Thoughts (PoT) prompting approach that improves the numerical reasoning capability of LLMs on financial datasets, including FinQA and ConvFinQA (Chen et al., 2022). PoT explicitly prompts the model to frame its calculations as a program, using programming languages as tooling. On TAT-QA, despite better performance against other prompting strategies such as CoT, PoT prompting falls short of the SotA performance. An error analysis reveals that the majority of errors are due to incorrect retrieval. This may be the result of the complex structure of tabular data in the TAT-QA dataset, which does not include standardized tabular structures. In (Wang et al., 2024a), the authors show that using equations (rather than programs) as intermediate meaning representations can

enhance the numeric reasoning capability of LLMs. By decomposing the prompts into sub-prompts that correspond to single equations, the authors demonstrate that the equations can better follow the logical order of calculations implied in the prompts, as opposed to programs, which have certain ordering constraints (e.g. a variable cannot be mentioned prior to being defined). Their equation-as-intermediate-meaning method, known as BRIDGE, outperforms other methods, including PoT, on math word problems. In (Zhu et al., 2024), the authors introduce TAT-LLM, a specialized LLM based on the Llama2-7B base (Touvron et al., 2023b) that can perform quantitative reasoning over text/tabular data. The authors instruction-tune the base model using a step-wise strategy that prompts the model to retrieve relevant evidence from the context, generate the reasoning steps, and produce the final answer accordingly. TAT-LLM outperforms SotA LLMs as well as SotA fine-tuned models on FinQA, TAT-QA, and TAT-DQA datasets. It improves the exact-match accuracy over the best-performing baseline (GPT-4) by an average of 2.8%.

In (Srivastava et al., 2024), the authors analyze the performance of LLMs on quantitative reasoning tasks over text/tabular contexts. They identify three common failure modes: 1) incorrect extraction of relevant evidence from the input, 2) incorrect generation of the reasoning program, and 3) incorrect execution of the program. An analysis of four financial QA datasets shows that reasoning and calculation errors dominate in datasets that provide standardized tabular data (FinQA (Chen et al., 2021b) and ConvFinQA (Chen et al., 2022)), whereas in datasets with complex tabular structures (TAT-QA (Zhu et al., 2021) and MultiHiertt (Zhao et al., 2022)), extraction errors are more common.

**Multimodal Understanding.** As noted above, the complex structure of tabular data can complicate numerical reasoning. Documents with visually rich content and complex layouts are common in financial domains, and studies such as (Ye et al., 2023a) and (Wang et al., 2023a) have demonstrated that the incorporation of visual and spatial features in the representation of text can enhance the performance of LLMs on tabular and visual reasoning tasks. In (Yue et al., 2024), the authors propose a framework for LLM-based information extraction from long documents containing hybrid text/tabular content. By serializing tabular data into text, dividing the documents into segments, retrieving relevant segments, and summarizing each retrieved segment, they show substantial improvement over basic prompt-based information extraction from financial documents. In (Ouyang et al., 2024), the authors demonstrate that the fusion of multimodal information (text, audio, video), with domain knowledge represented in a knowledge graph can lead to better predictions of the movement and volatility of financial assets. Notably, they use a Graph Convolutional Network (Kipf & Welling, 2017) as a universal fusion mechanism across modalities, and show the representations learned by the GCN can be used to instruction-tune an LLM to obtain superior performance to SotA approaches.

**LLMs & Time-Series Data.** Another data modality that is prominent in financial applications is time-series data. Research into time-series modeling has shown that pre-trained LLMs can be "patched" to model time-series data (Jin et al., 2024; Chang et al., 2024). Consistent with (Wu et al., 2023d), Gruver et al. (2023) demonstrate that number-aware tokenization enhances the performance of LLMs on time-series forecasting tasks, even in zero-shot settings. Yu et al. (2023e) demonstrate the ability of LLMs to perform explainable stock return prediction by combining time-series data with news and company metadata. The GPT-4 model, combined with CoT prompting outperforms other methods, including an instruction-tuned model based on Open LLaMA-13B.

**LLMs & Hallucination.** In (Kang & Liu, 2024), the authors analyze and profile the hallucination behaviors of SotA LLMs including GPT-4 and Llama-2, when applied to financial tasks. They demonstrate that a high rate of hallucination can occur with tasks that require financial domain knowledge or retrieval from pretraining data. They demonstrate that RAG can mitigate hallucinations on knowledge-sensitive tasks. For smaller LLMs, they recommend tuning and prompt-based tool learning as a mitigation strategy, such as data retrieval via APIs.

**Modeling Domain Knowledge.** A major challenge in financial applications is to fill the knowledge gap that exists between models trained on open-domain datasets, and tasks that require financial domain knowledge (Chen et al., 2021b; Aguda et al., 2024; Kang & Liu, 2024). Zhang et al. (2023b) demonstrate that instruction-tuning LLMs on domain-specific data and using RAG can enhance their performance on financial sentiment analysis. In (Deng et al., 2023b), the authors use the CoT (Wei et al., 2022) prompting to augment

an LLM with financial domain knowledge. The LLM is used to generate weak labels over social media posts, which are in turn used to train a small LM to detect market sentiment from social media. This method can be used to tackle several challenges in conventional approaches to market sentiment detection, including the scarcity of labeled data and the specificity of social media jargon. In (Zhao et al., 2023b), the authors introduce KnowledgeMath, a Math Word Problem benchmark for finance that requires college-level proficiency in the domain. They demonstrate that while knowledge augmentation techniques such as CoT (Wei et al., 2022) and PoT (Chen et al., 2023c) can enhance LLM performance by as much as 34%, the best-performing LLM achieves an overall score of 45.4, far from the human baseline of 94.

**LLM Agents.** The complexity of some tasks in the financial domain has motivated research into agent-based systems. In (Li et al., 2024b), the authors introduce FinMem, a multi-agent system for financial trading. The authors propose a multi-layered memory mechanism that helps LLM agents retrieve the most recent, relevant, and important events for a given trading decision. In addition, a profiling mechanism enables agents to emulate various trading personas and domains. When tested on five companies sampled from diverse trading sectors, FinMem yields significantly higher cumulative return compared to baselines. In (Park, 2024), the authors introduce a framework for anomaly detection in financial data using LLM agents. By casting agents as task experts, the author proposes a pipeline through which the agents undertake different sub-tasks such as data manipulation, detection, and verification. The output produced by these agents is then presented to "manager" agents, which will use an interactive debate mechanism (Li et al., 2023a) to derive the final output. Xing (2024) criticize the standard approach to the development of debating systems among homogeneous agents. Instead, the author proposes heterogeneous agents that can emulate roles or personas, arguing that a debate among heterogeneous agents can lead to better outcomes on semantically challenging tasks such as sentiment analysis.

### 3.5 Future Prospects

**Tasks and Datasets.** As a longstanding challenge for almost every interdisciplinary area, the knowledge gap between the NLP community and the finance community hinders the progress of financial NLP. This divide has led to the predominance of tasks focusing on shallow semantics and basic numerical reasoning, such as those illustrated by the FinQA dataset (Chen et al., 2021b), where questions typically require only rudimentary calculations like percentage changes. Such tasks, though valuable, are recognized as elementary within the realm of financial analysis (Li et al., 2023d). As shown in §3.3, current methods nearly reached saturated performance in relatively simple tasks like sentiment analysis and headline classification. In addition to the knowledge gap, another barrier is the difficulty and high cost of acquiring high-quality, real-world data. Apart from confidential data sources, creating financial datasets requires a high level of expertise. For example, the construction of the FinQA dataset (Chen et al., 2021b) involved the recruitment of eleven financial professionals for data annotation, which cost over 20k US dollars[9]. In the era of LLMs, the requirement for large-scale annotated training data is no longer a must, somewhat reducing the costs associated with data annotation. However, the expenses and logistical efforts involved in curating high-quality test data and maintaining productive collaborations with domain experts remain significant hurdles. These all require enough research funding, sufficient time commitment, and effective management. Here, we highlight some potential directions for future research:

- **Exploring Realistic Tasks.** Moving beyond surface-level tasks to embrace more sophisticated and realistic challenges is imperative. This involves the formulation of tasks that demand intricate reasoning as mentioned in §3.1, such as multi-step financial analyses, fraud detection, risk assessment, etc, that require building language agents (Wang et al., 2024c; Shinn et al., 2024; Sumers et al., 2023) capable of nuanced planning and decision-making processes in real-world financial analysis.

- **Incorporating Multimodal Documents.** Current financial NLP tasks still mostly target text or simple tables. Real-world financial documents may involve richer modalities and structures, such as complex tables with nested structures and charts of various structures. Understanding and performing financial reasoning on multiple modalities is under-explored.

---

[9]We contacted the authors of FinQA for this information.

- **Fostering Interdisciplinary Collaboration and Learning.** The development of high-fidelity financial NLP solutions necessitates closer collaboration between the NLP and finance sectors. In order to bridge the conceptual and methodological gaps between these fields, researchers should also take the initiative to acquire cross-disciplinary knowledge, with the goal of better understanding the imperative challenges in finance, as well as to achieve smoother and more effective communications with financial domain experts.

**Methodologies.** The development of LLM-based methods for financial tasks closely follows the general NLP community, from the early pre-training with downstream fine-tuning paradigm to the recent instruction fine-tuning paradigm. Task-specific fine-tuning could achieve good performance (Li et al., 2023d) but often incurs high costs. We believe that two major factors should still be emphasized when developing LLM-based methods for the financial domain: how to equip the LLMs with domain knowledge and reasoning skills, especially in a cost-effective setting. As shown in §3.3, current instruction fine-tuning approaches, especially relatively smaller models, still fall behind fine-tuning models or general LLMs in complex tasks. Though it is widely believed that large general LLMs will eventually lead to the best performance for broader domains, developing strong, lightweight domain-specific models is still a promising direction. Below are some potential future directions:

- **Knowledge-Intensive Instruction Fine-Tuning.** Beyond generic fine-tuning or instruction fine-tuning curated from existing datasets, we envision the development of novel paradigms that specifically enhance LLMs' understanding of complex financial concepts, terminologies, logics, and rules. This involves creating high-quality, finance-specific datasets for instruction fine-tuning that encapsulate the breadth and depth of the domain's knowledge.

- **RAG.** The RAG framework (Lewis et al., 2020) offers a compelling method for LLMs to dynamically integrate external domain knowledge into their generative processes. By adapting RAG for finance, LLMs can access and apply up-to-date market data, regulations, and financial theories, thereby enhancing their analytical and predictive capabilities.

**Deployment and Applications.** Despite the considerable amount of existing LLM research on finance, their real-world deployment remains scant. As suggested by (Li et al., 2023d), current LLMs excel primarily at straightforward financial NLP tasks; however, they falter when confronting more intricate challenges, failing to meet the rigorous standards of the industry. Given the high stakes of financial decision-making, where inaccuracies can precipitate substantial losses and legal entanglements, we believe the following dimensions are critical for transitioning from theoretical academia models to impactful real-world deployments:

- **Enhancing Accuracy and Robustness.** The systems cannot be deployed for real-world usages but are only limited to academic experiments until we reach a satisfactory level for industrial standards. Presently, academic benchmarks often lack the depth and realism to adequately prepare these models for practical tasks. Meanwhile, studying how to develop models that are robust to adversaries and attacks is also an important direction.

- **Evolving Human-AI Collaboration Paradigms.** How to design usage paradigms for real-world users is also an important future direction. Current systems predominantly operate under a paradigm of user assistance, augmenting rather than replacing the expertise of financial professionals (Chen et al., 2021b; 2022). We expect that more future works could be explored, such as advanced collaboration frameworks that enhance decision-making efficacy, system transparency, and user engagement, while also embedding HCI principles to foster intuitive and efficient user experiences.

- **Navigating Responsibility, Ethics, Regulations, and Legal Concerns.** The deployment of LLMs in high-stakes domains like finance necessitates a conscientious approach to design, underscored by ethical and legal foresight. Current work in academic settings rarely addresses these considerations in a comprehensive and systematic way. Issues such as fairness, accountability for misguided financial advice, and the ethical implications of AI-driven decision-making demand rigorous attention. Future developments must prioritize responsible AI frameworks that address these concerns head-on, ensuring that LLMs contribute positively and ethically to the financial ecosystem.

## 4 Medicine and Healthcare

NLP has made remarkable strides in the biomedical field, providing essential insights and capabilities for various healthcare and medical applications. The recent emergence of LLMs has brought significant advancements to the medical field, primarily by incorporating extensive medical knowledge during training. This section explores the impact of LLMs on diverse biomedical tasks, benchmarks, and real-world applications. It demonstrates not only the power of LLMs in the biomedical sphere but also highlights their potential in practical medical scenarios. The organization of this section is as follows: In §4.1, we give an overview of the tasks and benchmarks in the medical domain. In §4.2, we summarize the advance of LLMs in three aspects: (1) closed-source LLMs (e.g., GPT-4 (Achiam et al., 2023) and ChatGPT (OpenAI, 2022)) and their performance for medical applications; (2) open-sourced LLMs in the medical domain, including their training strategies, data, and performance; (3) multimodal medical LLMs that bridge natural language with other modalities and being applied beyond text-only tasks. In §4.3, §4.4, §4.5, and §4.6, we will delve into some of the practical applications of LLMs for clinical applications. We will present and discuss performance comparison of various task-specific methods and LLMs. Finally, in §4.7, we summarize our insights and discuss potential future directions.

### 4.1 Tasks and Benchmarks for Medical NLP

**Sentence Understanding** A fundamental task in clinical NLP is to process sentences and documents, which could help extract meaningful information from clinical documents and assist clinicians in decision-making processes. Dernoncourt & Lee (2017) proposed a dataset for sequential sentence classification, where sentences in medical abstracts are labeled with one of the following classes: background, objective, method, result, or conclusion, which can help researchers to skim through the literature more efficiently. Abnormality detection (Harzig et al., 2019; He et al., 2023c) aims to detect abnormal findings in clinical reports, with a similar goal to reduce the workload of radiologists. Ambiguity classification (He et al., 2023d) has a different purpose to focus on patient care, where it aims to find ambiguious sentences written by doctors, that could cause the misleading from patients.

**Clinical Information Extraction.** In the biomedical NLP community, a primary goal is the extraction of key variables from biomedical texts for effective biomedical text analysis. Clinical sense disambiguation interprets medical abbreviations within their clinical context into specific terminology, or conversely, translating medical terminology into abbreviations. This is particularly crucial for understanding clinical notes, which are frequently filled with complex jargon and abbreviations (He et al., 2023d). For example, the abbreviation 'pt' could mean patient, physical therapy, or prothrombin time, etc. This task is usually formatted as a multiple-choice problem and evaluated by accuracy and F1 scores. Biomedical evidence extraction focuses on automatically parsing clinical abstracts to extract key information, such as interventions and controls, from clinical trials, aiding the adoption of evidence-based medicine by synthesizing findings across research studies (Nye et al., 2018). Coreference resolution is essential for accurately identifying and linking noun phrases that refer to the same entity, such as a person or a medical term. This process is crucial in clinical contexts, where it helps to distinguish between a patient's own medical history and that of their family members (Zheng et al., 2011; Chen et al., 2021a). This task has been largely evaluated on the 2010 i2b2/VA challenge, which consists of thousands of coreference chains (Uzuner et al., 2010).

**Medical Question Answering.** Question answering (QA) in the medical domain is a fundamental task in NLP, requiring language models to answer particular questions based on their internal medical knowledge. This task not only demands a deep understanding of clinical terminologies and concepts but also requires the capability to comprehend and interpret complex medical reasoning given the question. Medical QA tasks are mainly formed as multiple-choice questions providing a set of possible answers for each question, from which the correct one must be chosen. This format is particularly useful for testing the language model's ability to discriminate between related concepts and to understand nuances in medical knowledge. MedQA(USMLE) (Jin et al., 2020) evaluates professional biomedical and clinical knowledge through 4-way multiple-choice questions from the US Medical Licensing Exam. MedMCQA (Pal et al., 2022) is a large scale 4-way multiple-choice dataset from Indian medical school entrance exams. HeadQA (Vilares & Gómez-Rodríguez, 2019) offers multiple-choice questions from specialized Spanish healthcare exams between

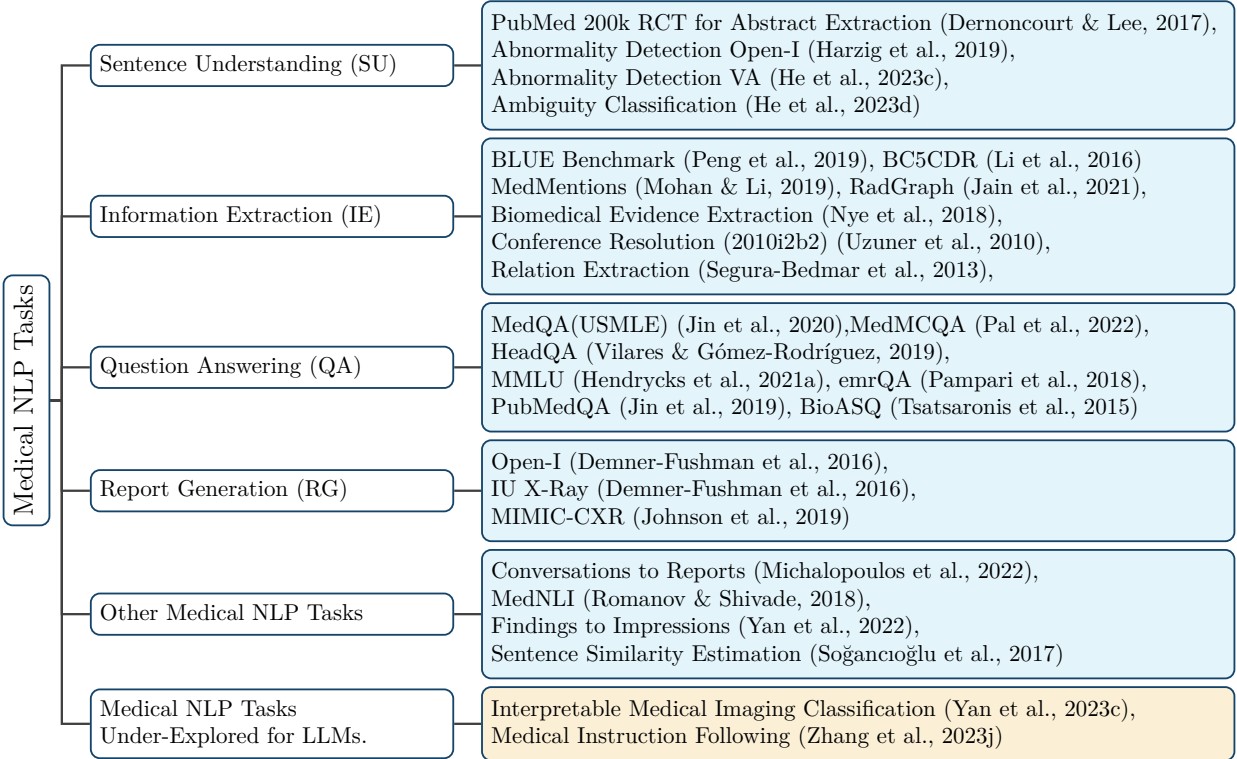

Figure 3: A summarization of medical NLP tasks and representative datasets. The yellow field shows the tasks relatively under-explored for LLMs.

2013 and 2017, with 2013 and 2014 featuring five-option tests and 2015 to 2017 having four-option tests. MMLU (Hendrycks et al., 2021a) includes a section of professional medicine questions with four-way multiple choices. PubMedQA (Jin et al., 2019) and BioASQ (Tsatsaronis et al., 2015) are reading comprehension datasets to answer yes/no/maybe based on a given passage.

In the following sections, we will discuss some of the representative tasks in the clinical setting, from abnormality detection and medical report generation, to some of the recently proposed tasks such as medical instruction evaluation and medical-imaging classification via natural language.

## 4.2 LLMs for Medicine and Healthcare

**Close-sourced Medical LLMs.** Close-sourced LLM pretraining for general proposes, such as ChatGPT (OpenAI, 2022) and GPT-4 (Achiam et al., 2023), have shown strong medical capacity across both medical benchmarks and real-world applications. Liévin et al. (2023) utilized GPT-3.5 with different prompting strategies, including CoT, few-shot, and retrieval augmentation, for three medical reasoning benchmarks, to show the model's strong medical reasoning ability in the absence of specific fine-tuning. The evaluation of LLMs, such as ChatGPT, on medical exams, including US Medical Exams (Kung et al., 2023) and Otolaryngology–Head and Neck Surgery Certification Examinations (Long et al., 2023), indicates that they achieve scores close to or at the passing threshold. This suggests the potential of LLMs to support real-world medical usages such as medical education and clinical decision-making. Agrawal et al. (2022) views LLMs such as GPT-3 as clinical information extractors and shows the potential in different information extraction tasks. The MedPaLM models (Singhal et al., 2022; 2023), are a series of medical domain-specific LLMs, adapted from PaLM models(?Anil et al., 2023; Chowdhery et al., 2022), which have shown performance in answering medical questions on par with that of medical professionals (Singhal et al., 2022; 2023). GPT-4 (Achiam et al., 2023) demonstrates strong medical capacities without specialized training strategies in the medical domain or engineering for solving clinical tasks (Nori et al., 2023a;b). When the scope is narrowed down to sub-domain domains, the performance of LLMs is variable. GPT-4 outperforms or performs on par with the

current SOTA radiology models (Liu et al., 2023b) across radiology tasks, and it matches human performance on gastroenterology board exam self-assessments (Ali et al., 2023). Peng et al. (2023) examines ChatGPT and GPT-4 performance on Physical Medicine and Rehabilitation, and demonstrates their potential capabilities in the submedical field. However, in dementia, LLMs fail to surpass the performance of traditional AI tools (Wang et al., 2023f). Besides directly utilizing LLMs for different medical tasks, they also can be used as a knowledge base for "retrieving" informative context to auxiliary downstream tasks (Yu et al., 2023d). For example, Zhang et al. (2023h) employs ChatGPT as a medical knowledge base to generate medical knowledge for supporting downstream medical decision-making. Kwon et al. (2024) generates clinical rationales for patient descriptions and utilizes the rationale as an additional training signal to fine-tune student models in both unimodal and multimodal settings to improve diagnosis prediction performance.

**Open-Sourced Medical LLMs.** Due to privacy concerns (Zhang et al., 2023h) and high costs, several open-source medical LLMs (Xu et al., 2023; Han et al., 2023; Li et al., 2023i; Wu et al., 2023b; Zhang et al., 2023j) have be built by tuning open-source base model, such as LLaMA (Touvron et al., 2023a;b), on medical corpus. These works mainly employ two different strategies: 1) continue pretraining followed by instruction-finetuning (Wu et al., 2023b; Xie et al., 2024a) and 2) direct instruction-finetuning (Xu et al., 2023; Han et al., 2023; Li et al., 2023i; Zhang et al., 2023j; Tran et al., 2023), as shown in 3. Specifically, the first approach involves continuously pretraining language models on biomedical corpora, including medical academic papers and textbooks, and then further fine-tuning the models with various medical instructional datasets to align with human intent for medical applications. The second approach directly conducts instruction fine-tuning on the base models to elicits the medical capabilities of base models directly. In the pretraining and fine-tuning schema, PMC-LLAMA (Wu et al., 2023b) employs a two-step training process that first extends LLaMA's training with millions of medical textbooks and papers, and then instructionally tunes the model on a dataset of 202 million tokens. Me LLaMA (Xie et al., 2024a) proposes a domain-specific base model by continuing to pretrain LLaMA-2 models with 13B and 70B parameters on a 129 billion token medical dataset, and then creates corresponding chat models by conducting instruction fine-tuning on 219k instances. On the other hand, ChatDoctor (Li et al., 2023i) collects 100k real online patient-doctor conversations and directly fine-tunes LLaMA on the dialogues dataset. MedAlpaca (Han et al., 2023) increases the instructional dataset to 230k including question-answer pairs and dialogues and conducts the fine-tuning procedure. Baize-Healthcare (Xu et al., 2023) employs about 100k dialogues from Quora and MedQuAD for instructional tuning by LoRA (Hu et al., 2022). AlpaCare (Zhang et al., 2023j) proposes a 52k diverse machine-generated medical instructional dataset, by distilling medical knowledge from robust closed-sourced LLMs (Li et al., 2022b). Then they fine-tune open-sourced LLMs on the dataset to show the importance of training data diversity on the model's ability to follow medical instructions while maintaining generalizability. BioInstruct (Tran et al., 2023) utilize GPT-4 to generate a 25k instruction dataset covering the topic in question-answering, information extraction and text generation to instruction tune LLaMA models (Touvron et al., 2023a;b) with LoRA (Hu et al., 2022). Their experimental results show consistent improvement in different medical NLP tasks compared to LLMs without instruction fine-tuning.

**Multimodal Medical LLMs.** Though LLMs have the potential to process and assist clinical NLP tasks, multimodal data (e.g., X-ray Radiology, CT, MRI, ultrasound) plays a vital role in medical and healthcare applications. When making diagnosis and medical suggestions, it is important for the model to have access to and being capable of understanding clinical modalities beyond text. Hence, there is a strong need to build multi-modal LLMs (Zhu et al., 2022; Liu et al., 2024b; Yan et al., 2023a; 2024a) that are capable of connecting language with other modalities. A representative open-domain model is GPT-4V (Achiam et al., 2023), where researchers have explored its potential for understanding X-rays (Yang et al., 2023b; Wu et al., 2023a). LLaVA-Med (Li et al., 2024a) leverages a figure-caption dataset extracted from PubMed Central, and uses GPT-4 to self-instruct open-ended instruction-following data from the captions to train a visual AI assistant. Gao et al. (2023a) uses a similar recipe to train a multimodal LLM for ophthalmology. Zhang et al. (2023d) trained their model with image-only, text-only, and multi-modal tasks like image captioning and VQA for radiology tasks.

Other than medical chatbots, another important aspect of multimodal medical LLMs is to transform other modalities into language space. To this end, various CLIP-style models (Radford et al., 2021b; Zhang et al., 2022; Wang et al., 2022; Bannur et al., 2023) with two stream visual and text encoder have been proposed.

| Model Name | Model Architecture | Training Corpus | Size |
|---|---|---|---|
| MedAlpaca (Han et al., 2023) | LLaMA | QA rephrased from wikidoc & Flashcards, Stackexchange medical Q&A | 7B,13B |
| ChatDoctor (Li et al., 2023i) | LLaMA | real patient-doctor conversations | 7B |
| Doctor GLM (Xiong et al., 2023) | GLM | medical Dialogues | 6.2B |
| Baize-Healthcare (Xu et al., 2023) | LLaMA | dialogues from Quora and MedQuAD | 7B |
| AlpaCare (Zhang et al., 2023j) | LLaMA | Alpaca data, ChatGPT & GPT-4 generated instruction data | 7B, 13B |
| BioInstruct (Tran et al., 2023) | LLaMA & LLaMA-2 | GPT-4 generated instruction data | 7B, 13B |
| Clinical Camel (Toma et al., 2023) | LLaMA-2 | Dialogues, articles, Medical QA | 13B, 70B |
| BioMedGPT (Luo et al., 2023) | LLaMA-2 | BioMed Articles, PubChemQA, UniProtQA | 7B, 10B |
| Meditron (Chen et al., 2023d) | LLaMA-2 | PubMed articles, abstracts, medical guidelines | 70B |
| PMC-LLAMA (Wu et al., 2024) | LLaMA | biomedical academic papers , textbook, medical QA, rationales, dialogues | 7B, 13B |
| Me LLaMA (Xie et al., 2024a) | LLaMA | medical data, medical instruction samples | 13B, 70B |
| BioMistral (Labrak et al., 2024) | Mistral | PubMed Central | 7B |
| LLaVA-Med (Li et al., 2024a) | LLaVA | figure-caption data from PubMed Central GPT-4 generated instruction data | 7B, 13B |

Table 3: Summary of open-sourced medical LLMs.

A downstream application is interpretable medical image classification (Yan et al., 2023c), which tries to generate medical concepts with LLMs and concept bottleneck models (Yan et al., 2023b; Echterhoff et al., 2024). This line of work leverages language to explain model decisions while also being able to keep similar or even better classification performance than black-box vision models.

## 4.3 Abnormality and Ambiguity Detection

Abnormality detection (Harzig et al., 2019) aims to identify abnormal findings in a radiology report by classifying if a sentence reports normal or abnormal conditions. In this task, language models are used to automatically read medical reports and reduce the workload of doctors.

Ambiguity detection was first proposed in (He et al., 2023d), where it tries to detect ambiguous sentences appear in radiology reports that lead to mis-interpretation of reports. Accurate identification of such sentences is crucial, as they impede patients' comprehension of diagnostic decisions and may cause potential treatment delays and irreparable consequences. As a novel task proposed recently, existing LMs may not readily include such a task into its pre-training stage. Therefore, evaluation of this task allows us to investigate how language models perform for unseen tasks.

Both tasks (He et al., 2023c) are sentence-level classification tasks. For comparison, we measured the classification performance of finetuned LMs (BERT (Devlin et al., 2019), RadBERT (Yan et al., 2022), BioBERT (Lee et al., 2020), ClinicalBERT (Huang et al., 2019), BlueBERT (Peng et al., 2019), BioMed-ReBERTa (Gururangan et al., 2020)) and prompted LLMs (GPT-3, ChatGPT, Vicuna, BioMed LM (Bolton et al., 2024)) by reporting their F1 scores, as shown in Table 4. One can observe that though general LLMs can make reasonable predictions and can improve their performance via few-shot learning, there is still a gap between finetuned LMs and prompted LLMs. Moreover, the novel task of ambiguity detection indeed raises challenges, and there is a need to improve the generalizabitily of LLMs to deal with unseen tasks.

| Models | | Chest | | Miscellaneous Domains | |
|---|---|---|---|---|---|
| | | Abnormality ↑ | Ambiguity ↑ | Abnormality↑ | Ambiguity ↑ |
| Fine-tuned LMs with task specific data | BERT | 0.9791 | **0.9893** | 0.9607 | 0.9749 |
| | RadBERT | 0.9794 | 0.9869 | 0.9640 | **0.9813** |
| | BioBERT | 0.9791 | 0.9862 | 0.9614 | 0.9743 |
| | ClinicalBERT | **0.9809** | 0.9874 | 0.9588 | 0.9736 |
| | BlueBERT | 0.9803 | 0.9867 | 0.9601 | 0.9775 |
| | BioMed-ReBERTa | 0.9569 | 0.9758 | **0.9776** | 0.9788 |
| Prompted LLMs with zero/few shot learning | zero-shot ChatGPT | 0.9277 | 0.6584 | 0.8880 | 0.5206 |
| | few-shot ChatGPT | 0.9498 | 0.5831 | 0.9099 | 0.5354 |
| | zero-shot GPT-3 | 0.8762 | 0.8742 | 0.8243 | 0.6448 |
| | few-shot GPT-3 | 0.9215 | 0.8320 | 0.9054 | 0.6371 |
| | zero-shot Vicuna-7B | 0.6987 | 0.2130 | 0.7261 | 0.3739 |
| | few-shot Vicuna-7B | 0.8071 | 0.0785 | 0.8166 | 0.2844 |
| | zero-shot BioMed LM | 0.6679 | 0.3485 | 0.6273 | 0.3726 |
| | few-shot BioMed LM | 0.7905 | 0.6804 | 0.7638 | 0.6804 |

Table 4: Evaluation (accuracy) over two categories of PLMs on abnormality identification and ambiguity identification tasks (sentence-level NLU). **Bold**: the highest performance. Underlined: the lowest. Results are sourced from (He et al., 2023c). Few-shot results used 5-shots.

## 4.4 Medical Report Generation

Medical report generation (Yan et al., 2021) aims to build models that take medical imaging studies (e.g., X-rays) as input and automatically generate informative medical reports. Unlike conventional image captioning benchmarks (e.g. MS-COCO (Lin et al., 2014)) where referenced captions are usually short, radiology reports are much longer with multiple sentences, which pose higher requirements for information selection, relation extraction, and content ordering. To generate informative text from a radiology image study, a caption model is required to understand the content, identify abnormal positions in an image and organize the wording to describe findings in images.

Evaluation of this task involve two aspects: (1) Automatic metrics for natural language generation: BLEU (Papineni et al., 2002), ROUGE-L (Lin, 2004), and METEOR (Denkowski & Lavie, 2011). (2) Clinical Efficiency: CheXpert labeler (Irvin et al., 2019) is used to evaluate the clinical accuracy of the abnormal findings reported by each model, which is a state-of-the-art rule-based chest X-ray report labeling system (Irvin et al., 2019). Given sentences of abnormal findings, CheXpert will give a positive and negative label for 14 diseases. We can then calculate the Precision, Recall and Accuracy for each disease based on the labels obtained from each model's output and from the ground-truth reports.

We report performance of representative clinical models and recent LLMs in Table 5. We consider the following baslines: *ST* (Xu et al., 2015), $M^2$ *Trans* (Miura et al., 2020), *R2Gen* (Chen et al., 2020), *WCL* (Yan et al., 2021), as well as recent work that uses LLMs: XrayGPT (Thawkar et al., 2023), RaDialog (Pellegrini et al., 2023), Rad-MiniGPT-4 (Liu et al., 2024a). We observe similar trend as the sentence classification tasks, even though LLMs are good at generating fluent text and achieving high NLG scores, domain-specific models can still outperform LLMs in terms of clinical efficacy.

## 4.5 Medical Free-form Instruction Evaluation

Free-form instruction evaluations assess the practical medical value of language models from a user-centric perspective. This task involves inputting a medical query in a free-text format into the model, which then generates a corresponding response. For instance, if a user inputs, 'Discuss the four major types of leukocytes and their roles in the human immune system in bullet point format,' the model will produce an informed answer based on its internal medical knowledge. This task serves to measure both the medical knowledge capacity and the instruction-following capability of the model. iCliniq (Li et al., 2023i; Chen et al., 2024) contains 10k real online conversations between patients and doctors to evaluate models' medical instruction-following ability in the dialog scenario. MedInstruct-test (Zhang et al., 2023j) contains 217 clinical

| Model | NLG metrics | | | | CE metrics | | |
|---|---|---|---|---|---|---|---|
| | BLEU-1 | BLEU-4 | METEOR | ROUGE-L | Precision | Recall | F-1 |
| ST | 29.9 | 8.4 | 12.4 | 26.3 | 24.9 | 20.3 | 20.4 |
| $M^2$ Trans | - | 11.4 | - | - | **50.3** | **65.1** | **56.7** |
| R2Gen | 35.3 | 10.3 | 14.2 | 27.7 | 33.3 | 27.3 | 27.6 |
| WCL | 37.3 | 10.7 | 14.4 | 27.4 | 38.5 | 27.4 | 29.4 |
| XrayGPT | - | - | - | 20.0 | - | - | - |
| RaDialog | 34.6 | 9.5 | 14.0 | 27.1 | - | - | 39.4 |
| Rad-MiniGPT-4 | **40.2** | **12.8** | **17.5** | **29.1** | 46.5 | 48.2 | 47.3 |

Table 5: Performance comparison on the test set of MIMIC-CXR with respect to natural language generation (NLG) and clinical efficacy (CE) metrics. Results are reported in percentage (%).

| | iCliniq | | | | | MedInstruct | | | | |
|---|---|---|---|---|---|---|---|---|---|---|
| | Text-davinci-003 | GPT-3.5-turbo | GPT-4 | Claude-2 | AVG | Text-davinci-003 | GPT-3.5-turbo | GPT-4 | Claude-2 | AVG |
| Alpaca | 38.8 | 30.4 | 12.8 | 15.6 | 24.4 | 25.0 | 20.6 | 21.5 | 15.6 | 22.5 |
| ChatDoctor | 25.4 | 16.7 | 6.5 | 9.3 | 14.5 | 35.6 | 18.3 | 20.4 | 13.4 | 18.2 |
| Medalpaca | 35.6 | 24.3 | 10.1 | 13.2 | 20.8 | 45.1 | 33.5 | 34.0 | 29.2 | 28.1 |
| PMC | 8.3 | 7.2 | 6.5 | 0.2 | 5.5 | 5.1 | 4.5 | 4.6 | 0.2 | 4.6 |
| Baize-H | 41.8 | 36.3 | 19.2 | 20.6 | 29.5 | 35.1 | 22.2 | 22.2 | 15.6 | 26.6 |
| AlpaCare | **66.6** | **50.6** | **47.4** | **49.7** | **67.6** | **53.6** | **49.8** | **48.1** | **48.4** | **53.5** |

Table 6: **Performance comparison of medical LLMs on medical free-form instruction evaluation.** GPT-3.5-turbo acts as a judge for pairwise auto-evaluation. Each instruction-tuned model is compared with 4 distinct reference models: Text-davinci-003, GPT-3.5-turbo, GPT-4, and Claude-2. 'AVG' denotes the average performance score across all referenced models in each test set. The table is sourced from (Zhang et al., 2023j).

craft free-form instructions to evaluate the medical capacity and instruction-following ability of models across different medical settings such as treatment recommendation, medical education, disease classification, etc. Evaluating the instruction-following capacity of LLMs is complex due to the wide range of valid responses to a single instruction and the difficulty of replicating human evaluations. Recently, automated evaluation (Zheng et al., 2023; Dubois et al., 2023; Zhang et al., 2023i; Lu et al., 2022) has offered greater scalability and explainability compared to human studies. A strong LLM is used as a judge to compare the outputs of the evaluated model with reference answers and then calculate the winning rate of the evaluated model against the reference answer is used as the evaluation metric. Table 6 shows the current open-source medical LLM performance on medical free-form instruction evaluation with GPT-3.5-turbo acts as a judge and Text-davinci-003, GPT-3.5-turbo, GPT-4, and Claude-2 as reference models, respectively.

## 4.6 Medical-Imaging Classification Via Natural Language

Medical imaging classification with deep learning has long been studied in the computer vision and clinical community (Li et al., 2014). The task asks a model to take medical imaging (e.g., CT scans) as input, and assign diagnostic labels to them. However, predicting medical symptoms with "black-box" deep neural models could raise safety and trust issues, as it is hard for human to understand model behaviors and trust model decisions at ease. Clinicians often need to understand the underlying reasoning of the models to carefully make their decisions. Interpretable models allow for better error analysis, bias detection, ensuring patients safety, and trust building. Most recently, the idea of concept bottleneck models (CBMs) (Koh et al., 2020) has been introduced to medical imaging classification, where one can build an intermediate layer by projecting latent image features into a concept space to bring interpretability in the form of natural language.

A follow-up work (Yan et al., 2023c) further shows that classification with concepts not only bring interpretability, but also offers robustness, with the help of pretrained multi-modal LMs (an illustration is presented

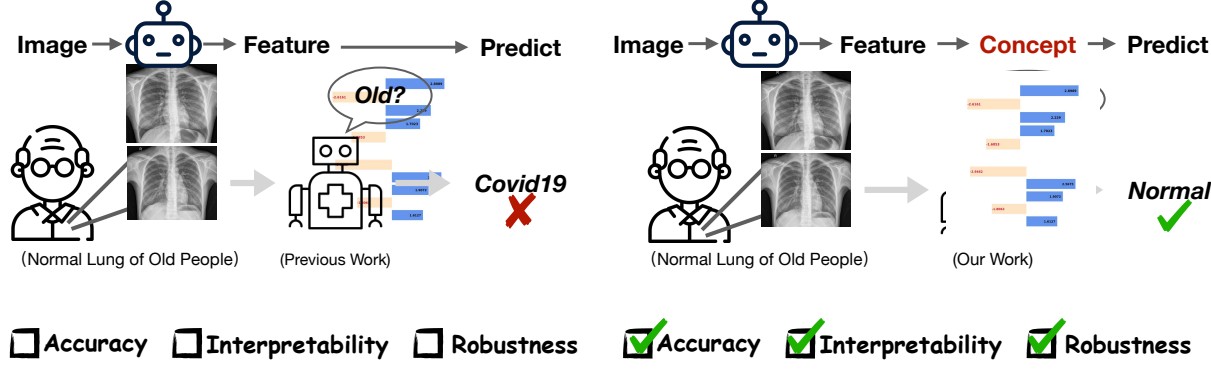

Figure 4: High-level illustration of concept bottleneck models (Yan et al., 2023c). It uses concepts for medical image classification to achieve interpretability and robustness while maintaining accuracy. **Left**: Classification with a classical neural encoder; **Right**: Classification with natural language concepts. A Chest X-ray from a healthy old individual may be classified as Covid-19 due to the patient's age, while introducing language can mitigate the effect of these confounding factors.

in Figure 4). This is especially important to medical applications, as confounding factors broadly exist and labeled data are often limited (De Bruijne, 2016). Take the classification of patient X-rays between Covid-19 and normal for instance, certain factors such as the hospitals where the X-rays are performed and the age of the patient strongly correlate with the target disease classification. Yan et al. (2023c) created four diagnostic benchmarks with different confounding factors: age, gender, hospital system. Here we present results on these datasets with comparison for (1) state-of-the-art robust machine learning methods: ERM and Fish (Shi et al., 2021), Lisa (Yao et al., 2022b); (2) linear probing on image features; (3) CBMs with different vision-language models as the backbone: CLIP (Radford et al., 2021b), MedCLIP (Wang et al., 2022), and BioViL (Bannur et al., 2023), shown in Table 7. We find that BioVIL shows promising results when evaluating on challenging datasets with various confounding factors, while another medical VLM, MedCLIP performs similar to general domain CLIP model. To enable useful concept bottleneck models, a strong and robust domain-specific vision-language model is needed.

| Models | NIH-gender | NIH-age | NIH-agemix | Covid-mix | Interpretability |
|---|---|---|---|---|---|
| ERM | 21.70 | 3.30 | 13.80 | 51.73 | ✗ |
| Fish | 21.70 | 6.00 | 17.00 | 52.16 | ✗ |
| LISA | 23.00 | 2.30 | 14.20 | 51.30 | ✗ |
| BioViL Image Features | 71.60 | 9.40 | 13.70 | 51.08 | ✗ |
| BioViL Image Features (dropouts) | 70.20 | 19.00 | 28.60 | 49.57 | ✗ |
| CBM w/ CLIP | 35.80 | 15.00 | 19.10 | **64.93** | ✓ |
| CBM w/ MedCLIP | 42.00 | 20.10 | 21.10 | 51.95 | ✓ |
| CBM w/ BioViL | **79.60** | **50.70** | **53.40** | 62.36 | ✓ |

Table 7: Performance comparison for robust medical imaging classification. Results are reported in accuracy (%) and sourced from (Yan et al., 2023c).

## 4.7 Future Prospects

**Improving the Capacity of Open-sourced Medical LLMs.** Open-source, domain-specific medical LLMs aim to narrow the performance gap between powerful closed-source LLMs and enhance the capability of smaller models to follow various medical instructions and align with user intentions by conducting continuous

pretraining and instructional fine-tuning. To further improve the capacity of these models, several future directions can be considered:

- **Data Diversity and Quality.** Although machine-generated datasets accelerate data generation for LLM training, their diversity still lags behind that of real-world collected datasets, which highly impacts model performance (Chiang et al., 2023). Expanding the training datasets to include a broader range of real-world medical texts, such as clinical trial reports, medical journals, patient records, and health forums, can improve the model's understanding of diverse medical contexts and terminologies. Additionally, ensuring the quality and reliability of the training data is crucial for maintaining the accuracy and trustworthiness of the model's outputs.

- **RAG.** Integrating RAG techniques can enhance the model's ability to access and incorporate relevant medical knowledge from extensive sources such as large medical knowledge bases, private hospital records, and databases. This approach can improve the model's responses by providing more accurate and contextually appropriate information, particularly in complex medical scenarios in inference time.

- **Addressing Privacy Concerns.** Medical data usage has strong restrictions compared to the general domain and other small domains. Developing methodologies address privacy issues in utilizing LLM APIs and building local LLMs are both important. This can include implementing secure data transmission protocols, ensuring data anonymization, and adopting privacy-preserving techniques such as differential privacy.

**Learning in a Data Sparsity Setting.** A critical challenge in medical domain for training large-scale models is the restriction of data usage. Data sparsity is a persistent issue due to privacy and confidentiality concerns, the cost of data Acquisition and annotation, as well as ethical considerations. For many practical tasks, e.g., medical report generation, clinical chatbots, medical image classification, the data sparsity issue will be a remaining challenge. Additionally, as mentioned earlier in the performance comparison sections for different applications, task-specific models that trained with in-distribution data and specific architectural design can still outperform foundation models. Based on the empirical finding of training general domain LLMs (Brown et al., 2020; Kaplan et al., 2020; Achiam et al., 2023), scaling up data is of vital importance for model performance. We discuss some of the potential future directions to address this issue:

- **Transfer Learning and Domain Adaptation.** For medical LLMs, it is worth exploring how scaling up general domain, publicly-available data can help with in-domain medical tasks. We can explore data selection strategies in pretraining stage to improve the transfer learning performance from general-domain models to medical-specific tasks.

- **Synthetic Data Generation.** To mitigate the challenges posed by data scarcity, another approach is the generation of synthetic medical data. Leveraging advanced LLMs could enable the creation of diverse synthetic datasets to augment the learning process.

- **Few-shot and Zero-shot Learning.** Few-shot learning and in-context learning methods should be explored more deeply, which has the potential to let medical LLMs adapting to new tasks or domains with minimal training data.

- **Privacy-preserving Techniques.** Techniques such as differential privacy and federated learning (Rieke et al., 2020) could allow the utilization of patient data for training purposes while ensuring that individual privacy is maintained.

**Active Learning.** Implementing active learning strategies where the model identifies the most informative data points for labeling can optimize the training process. This method ensures efficient use of scarce data resources and improves learning outcomes in highly specialized medical contexts.

**Evaluating Real-world Medical Application Capacity.** Although various benchmarks have been proposed in the medical domain, most of them focus on evaluating models from the perspective of medical

knowledge (Pal et al., 2022; Jin et al., 2020; 2019), rather than from a user-oriented perspective. To bridge this gap, free-form instruction evaluation datasets utilize medical dialogues and machine-generated text for medical questions. However, these test sets are still limited in quantity and task diversity. For example, Med-Instruct proposes an evaluation dataset with diverse topics, but it is limited to only two hundred tests. On the other hand, iCliniq contains 10,000 instances, but its scope is restricted to doctor-patient conversations. Therefore, there is a need for a large-scale, diverse, and expert-verified dataset for evaluating the medical capacity of LLMs in real-world medical user applications. To evaluate the response quality of LLM in medical free-form instruction evaluation, Zhang et al. (2023j) utilizes the LLM APIs as judges (Zheng et al., 2023; Dubois et al., 2023; Zhang et al., 2023i). However, calling LLM APIs for evaluation is costly. Therefore, training a smaller LLM with strong medical capacity for response evaluation and comparison could be a more efficient alternative. Future work could focus on techniques such as knowledge distillation or model pruning to create such a medical specialized evaluator, potentially leading to faster and more cost-effective evaluation processes for medical LLM applications.

## 5 Law

NLP is pivotal in the legal domain, providing sophisticated tools for managing the extensive and intricate textual data inherent in legal documentation and proceedings (Harvard Law School Library, 2023; United States Congress, 2023; Fang et al., 2023). The advent of LLMs has further catalyzed innovation at the frontier of legal applications. This section explores the profound influence of LLMs across a range of legal tasks. These technological advancements have strengthened significant enhancements in areas such as legal judgment prediction, legal event detection, legal text classification, and legal document summarization. The purpose of this chapter is to outline the trajectory of LLMs in revolutionizing legal NLP and to shed light on both the challenges faced and the potential future developments. This chapter is organized as follows: In §5.1, we introduce the current NLP tasks in the legal domain, detailing task formulations and relevant datasets. In section §5.2, we explore various PLMs and LLMs developed specifically for legal applications. In section §5.3, we examine the evaluations and performance analysis of LLMs in legal contexts. In section §5.4, we discuss various LLM-based methodologies developed for tackling legal tasks and challenges. Finally, we summarize insights, draw conclusions, and discuss potential future directions in section §5.5.

### 5.1 Tasks and Datasets in Legal NLP

In this section, we present an array of legal tasks and corresponding datasets that have been investigated through the LLM methodologies. The domains covered include legal question answering (LQA), legal judgment prediction (LJP), legal event detection (LED), legal text classification (LTC), legal document summarization (LDS), and other NLP tasks. Figure 5 provides an overview of these established legal NLP tasks and related datasets.

**Legal Question Answering (LQA).** LQA is the process of providing answers to legal questions and promotes the development of systems proficient in handling complex inquiries related to laws, regulations, case precedents, and theoretical syntheses. The LQA dataset comprises a wide array of question-and-answer pairs that serve to evaluate a system's capability in legal reasoning. CRJC (Duan et al., 2019), akin to the SQUAD 2.0 (Rajpurkar et al., 2018) format, includes challenges such as span extraction, yes/no questions, and unanswerable questions. Furthermore, professional qualification examinations like the bar exam require specialized legal knowledge and skills, making datasets such as the MBE (Wyner et al., 2016) from the US, JEC-QA (Zhong et al., 2020) from China, and COLIEE2015 (Kim et al., 2015) from Japan particularly demanding. Specific legal domains also have dedicated datasets. For instance, SARA (Holzenberger et al., 2020) focuses on US tax law and includes test cases, while VLQA (Bach et al., 2017) addresses Vietnamese transportation law. In the area of privacy law, PrivacyQA (Ahmad et al., 2020) and PIL (Sovrano et al., 2021) test the system's ability to navigate complex language and regulations regarding data privacy. For the community-oriented legal education, FALQU (Mansouri & Campos, 2023) and LCQA (Askari et al., 2022) are obtained from Law Stack Exchange (law). Several databases employ specific techniques to improve the datasets' quality, for example, EQUALS (Chen et al., 2023a) filters out unqualified legal questions from the raw data. AILA (Huang et al., 2020) integrates domain knowledge from a legal knowledge graph to

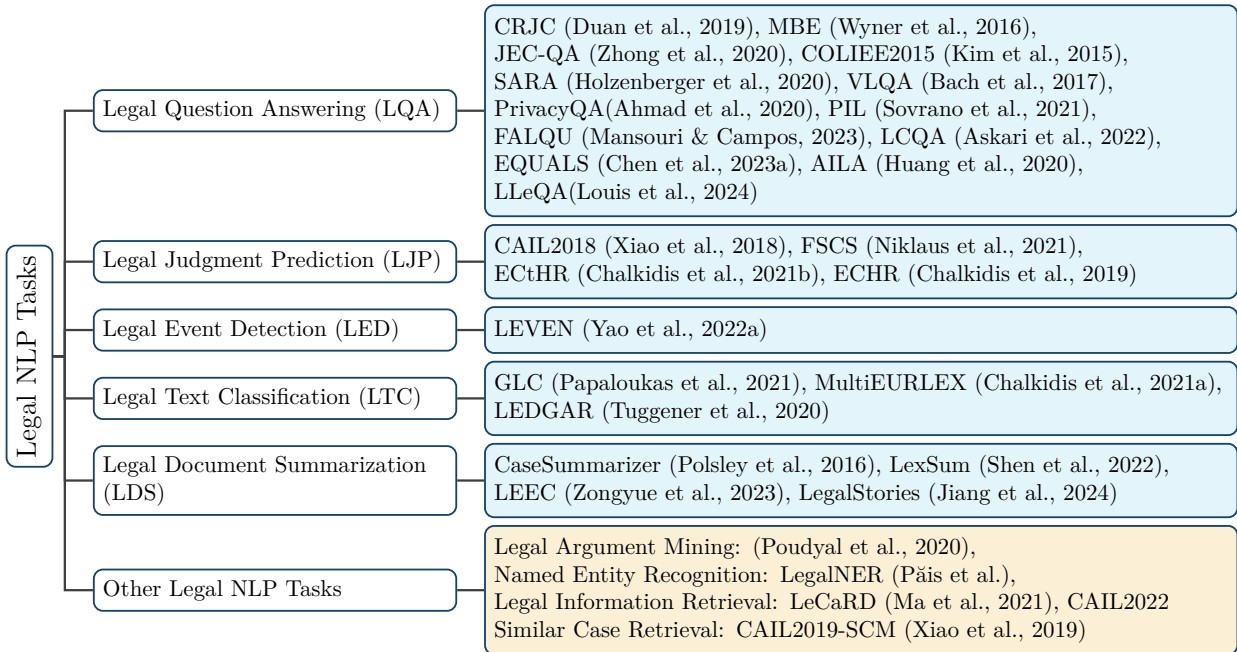

Figure 5: A summarization of existing legal NLP tasks and datasets. The yellow field shows other legal tasks.

comprehend and rank question-answer pairs effectively. LLeQA (Louis et al., 2024) provides long-form answers to statutory law questions using a retrieve-then-read pipeline.

**Legal Judgment Prediction (LJP).** LJP focuses on analyzing legal texts such as case law, statutes, and trial transcripts to predict the outcomes of legal cases. This can assist judges, lawyers, and legal scholars in understanding potential case outcomes based on historical data. The task is generally treated as a classification problem where the input is a legal document and the target is a legal decision (e.g., conviction, acquittal, liability). Researchers have developed several datasets tailored to different legal systems across the globe. For instance, CAIL2018 (Xiao et al., 2018) is a comprehensive Chinese criminal judgment prediction dataset comprising over 2.68 million legal documents published by the Chinese government. Similarly, in Europe, datasets such as FSCS (Niklaus et al., 2021) offer insights into Swiss court judgments with 85,000 cases across two outcomes, reflecting the multilingual nature of the Swiss legal environment. The ECtHR (Chalkidis et al., 2021b) and ECHR (Chalkidis et al., 2019) datasets focus on European Union court judgments, each containing around 11,000 cases but offering a broader scope with 11 potential outcomes.

**Legal Event Detection (LED).** LED in legal documents involves identifying significant legal proceedings or decisions, such as rulings, motions, or amendments. This task is crucial for enabling legal professionals to monitor pivotal developments within cases efficiently. While Shen et al. (2020) propose hierarchical event features to distinguish similar events in legal texts, and Li et al. (2020b) implement event extraction technologies specifically for the description segments of Chinese legal texts, these studies are constrained by their datasets, which contain only thousands of event mentions. Such limited annotations fail to provide robust training signals or reliable evaluation benchmarks. Addressing this gap, LEVEN (Yao et al., 2022a), a comprehensive and high-quality dataset, is designed to enhance the capabilities of legal information extraction and LED.

**Legal Text Classification (LTC).** LTC involves categorizing structured sections within legal documents to enhance their accessibility and comprehensibility. For instance, most legal documents contain sections like "Facts of the case," "Arguments presented by the parties," and "Decisions of the current court," whose identification is crucial for understanding the legal outcomes of cases. These documents can thus be categorized into classes such as facts, argument, and statute, making LTC a multi-class classification task. Key datasets that have propelled advancements in LTC include the following: Greek Legal Code(GLC) (Papaloukas et al., 2021) focuses on categorizing a wide array of Greek legal documents; MultiEURLEX (Chalkidis et al., 2021a)

provides a broad collection of EU legislation for classification across multiple languages and jurisdictions; LEDGAR (Tuggener et al., 2020) datasets include large collections of contracts, offering detailed classification based on contract elements and terms.

**Legal Document Summarization (LDS).** LDS aims at condensing legal documents into succinct summaries while preserving key legal arguments and outcomes. The CaseSummarizer (Polsley et al., 2016) dataset focuses on summarizing case judgments, providing a concise overview of case facts, legal arguments, and judgments. Another dataset, LexSum (Shen et al., 2022), targets the summarization of legislative texts, aiming to extract essential elements and implications for easier comprehension. LEEC (Zongyue et al., 2023) is a comprehensive, large-scale criminal element extraction dataset with 15,831 judicial documents and 159 labels to address the limitations of existing datasets in legal knowledge extraction. LegalStories (Jiang et al., 2024) has 295 complex legal doctrines, each paired with a story and multiple-choice questions generated by LLMs.

**Other Legal NLP Tasks.** In recent developments, a couple of other tasks have emerged. Among them, legal argument mining (Poudyal et al., 2020) aims to detect and classify arguments within legal texts. Information extraction in the legal domain involves identifying and categorizing key legal entities, such as party names, locations, legal citations, and case facts. LegalNER (Păis et al.) is a dataset for extracting named entities from legal decisions. LeCaRD (Ma et al., 2021) and CAIL2022 datasets (Competition, 2022) enhance criminal case retrieval in Chinese law by linking fact paragraphs to full cases. Another emerging task is similar case retrieval, which aims to identify legal precedents and analogous cases to aid in legal decision-making. The CAIL2019-SCM dataset (Xiao et al., 2019), containing 8,964 triplets of cases published by the Supreme People's Court of China, underscores this task by focusing on the detection of similar cases. These tasks collectively enrich the technological landscape and hold promise for significant enhancements in the efficiency, accessibility, and fairness of legal services.

## 5.2 Legal LLMs

Since the development of BERT (Devlin et al., 2019), there have been continuous efforts to build PLMs and LLMs specialized for the legal domain. Following the evolving paradigms of general PLMs and LLMs, early legal PLMs adopted the pre-training followed by downstream task fine-tuning paradigm and initially trained relatively small language models. Recent works have scaled up model sizes and introduced instruction fine-tuning, with evaluations covering a broader set of legal tasks. Most existing legal LLMs are text-based, with a focus on Chinese, English, or multi-language support. Table 8 summarizes the PLMs and LLMs for the legal domain.

**Pre-Trained and Fine-Tuning PLMs.** LegalBERT (Chalkidis et al., 2020) is an early attempt to build a legal PLM targeting tasks like LTC. The model is further pre-trained on a corpus of legal documents and then fine-tuned using task-specific data. Lawformer (Xiao et al., 2021), a transformer-based model, is pre-trained specifically for handling lengthy legal texts, aiding in tasks such as LJP, LRC, and LQA.

**Pre-Trained and Fine-Tuning LLMs.** Pre-trained and fine-tuning LLMs involve LLMs specifically trained and fine-tuned for legal tasks or datasets. These legal-specific LLMs often integrate external knowledge bases and process extensive initial training to handle a wide range of legal data. Recent developments have led to models like LexiLaw (Haitao, 2024), a fine-tuned Chinese legal model based on the ChatGLM-6B (Group, 2023), meanwhile Fuzi.mingcha (SDU, 2023) is also based on ChatGLM-6B (Group, 2023), which is fine-tuned on CAIL2018 (Xiao et al., 2018) and LaWGPT (Xiao-Song, 2024). Furthermore, WisdomInterrogatory (LLM, 2023) is a pre-trained and fine-tuning model built upon Baichuan-7B (Inc., 2023). More 7B LLMs like LawGPT-7B-beta1.0 (Nguyen, 2023) are pre-trained on 500k Chinese judgment documents upon Chinese-LLaMA-7B (Cui & et al., 2023), and HanFei (He et al., 2023b) is a fully pre-trained and fine-tuned LLM with 7B parameters. There are more explorations on large-scale LLMs, LaywerLLaM (Zhe, 2023) is based on Chinese-LLaMA-13B (Cui & et al., 2023), fine-tuned with general and legal instructions, additionally, ChatLaw-13B (Cui et al., 2023a) is fine-tuned based on Ziya-LLaMA-13B-v1 (IDEA-CCNL, 2023), and ChatLaw-33B (Cui et al., 2023a) is fine-tuned based on Anima-33B (Ogavinee & et al., 2022). It is worth noting that LLMs based on other languages have also recently emerged, such as SaulLM-7B (Colombo et al., 2024) based on Mistral-7B (Jiang et al., 2023) and JURU (Junior et al., 2024), which is the first LLM

| Model Name | Model Architecture | Main Evaluation Tasks | Languages | Size | Year |
|---|---|---|---|---|---|
| *Pre-trained and Downstream task Fine-tuning PLMs* | | | | | |
| LEGAL-BERT-SMALL (Chalkidis et al., 2020) | BERT (Devlin et al., 2019) | LTC, ST, NER | English | 35M | 2020 |
| LEGAL-BERT-BASE (SC) (Chalkidis et al., 2020) | BERT (Devlin et al., 2019) | LTC, ST, NER | English | 110M | 2020 |
| LEGAL-BERT-FP (Chalkidis et al., 2020) | BERT (Devlin et al., 2019) | LTC, ST, NER | English | 110M | 2020 |
| Lawformer (Xiao et al., 2021) | Longformer (Beltagy et al., 2020) | LJP, LRC, LQA | Chinese | 479M | 2021 |
| *Pre-trained Fine-Tuning LLMs* | | | | | |
| JURU (Junior et al., 2024) | Sabiá-2 (Sales Almeida et al., 2024) | LQA | Portuguese | 1.9B | 2024 |
| LexiLaw (Haitao, 2024) | ChatGLM-6B (Group, 2023) | LRC, LQA | Chinese | 6B | 2023 |
| Fuzi-Mingcha (SDU, 2023) | ChatGLM-6B (Group, 2023) | LJP, LRC, LQA | Chinese | 6B | 2023 |
| WisdomInterrogatory (LLM, 2023) | Baichuan-7B (Inc., 2023) | LJP, LRC, LQA | Chinese | 7B | 2023 |
| LawGPT-7B-beta1.0 (Xiao-Song, 2024) | Chinese-LLaMA-7B (Cui & et al., 2023) | LRC, LQA | Chinese | 7B | 2023 |
| SaulLM-7B (Colombo et al., 2024) | Mistral-7B (Jiang et al., 2023) | LQA | English | 7B | 2024 |
| Lawyer-LLaMA (Zhe, 2024) | Chinese-LLaMA-13B (Cui & et al., 2023) | LJP, LRC, LQA | Chinese | 13B | 2023 |
| ChatLaw-13B (Cui et al., 2023a) | Ziya-LLaMA-13B-v1 (IDEA-CCNL, 2023) | LJP, LRC, LQA | Chinese | 13B | 2023 |
| ChatLaw-33B (Cui et al., 2023a) | Anima-33B (Ogavinee & et al., 2022) | LJP, LRC, LQA | Chinese | 33B | 2023 |

Table 8: Summary of legal PLMs and LLMs. For evaluation tasks, we have **LTC** for Legal Text Classification, **ST** for Sequence Tagging, **NER** for Named Entity Recognition, **LJP** for Legal Judgment Prediction, **SCR** for Similar Case Retrieval, **LRC** for Legal Reading Comprehension, and **LQA** for Legal Question Answering.

| Model | JEC-QA | LEVEN | LawGPT | CAIL2018 |
|---|---|---|---|---|
| | Accuracy | F-1 | Rouge-L | F-1 |
| Fuzi-Mingcha (SDU, 2023) (zero-shot) | 0.08 | 0.17 | 0.22 | 0.25 |
| Fuzi-Mingcha (SDU, 2023) (few-shot) | 0.13 | 0.21 | 0.33 | 0.04 |
| ChatLaw-13B (Cui et al., 2023a) (zero-shot) | 0.28 | 0.32 | 0.31 | 0.33 |
| ChatLaw-13B (Cui et al., 2023a) (few-shot) | 0.29 | 0.40 | 0.34 | 0.26 |
| Wisdom-Interrogatory (LLM, 2023) (zero-shot) | 0.15 | 0.16 | 0.32 | 0.33 |
| Wisdom-Interrogatory (LLM, 2023) (few-shot) | 0.15 | 0.16 | 0.23 | 0.20 |
| GPT-3.5 (zero-shot) | 0.36 | 0.66 | 0.34 | 0.29 |
| GPT-3.5 (few-shot) | 0.37 | 0.68 | 0.52 | 0.31 |
| GPT-4 (zero-shot) | 0.55 | 0.79 | 0.33 | 0.52 |
| GPT-4 (few-shot) | 0.55 | 0.77 | 0.57 | 0.53 |

Table 9: Performance comparisons for LQA tasks (JEC-QA dataset (Zhong et al., 2020)), LED task (LEVEN dataset (Yao et al., 2022a)), and LJP task (LawGPT dataset (Xiao-Song, 2024) and CAIL2018 (Xiao et al., 2018)). We focus more on LJP tasks based on fact-based articles for the CAIL2018 dataset (Xiao et al., 2018) while scene-based articles for the LawGPT dataset (Xiao-Song, 2024). The few-shot setting is one shot for all datasets.

pre-trained for the Brazilian legal domain. These legal-specific LLMs, often following an initial pre-training phase, are tailored to specific legal datasets and tasks, enhancing both the precision and applicability of legal NLP technologies in practice.

## 5.3 Evaluation and Analysis of LLMs

The evaluation and analysis of LLMs' performance is crucial for understanding their effectiveness and capabilities, particularly in legal-specific contexts. This section introduces the evaluation benchmarks in assessing legal capabilities before the rise of LLMs. Subsequently, we explore specialized legal benchmarks designed explicitly for evaluating the performance of LLMs, and summarize their main findings. These works provide a focused and rigorous assessment of LLMs' abilities in handling legal tasks, offering insights into their efficacy and potential for legal applications.

Before the emergence of LLMs, there were benchmarks used to evaluate NLP models' legal performance. To evaluate model performance uniformly across diverse legal natural language understanding (NLU) tasks, LexGLUE benchmarks (Chalkidis et al., 2021c) are introduced. These benchmarks include datasets like ECtHR (Chalkidis et al., 2021b), SCOTUS (Spaeth et al., 2017), EUR-LEX (Chalkidis et al., 2021a),

LEDGAR (Tuggener et al., 2020), UNFAIR-ToS (Lippi et al., 2019), and CaseHOLD (Zheng et al., 2021). They provide a standardized framework for assessing the performance of the language models, allowing for systematic comparison and analysis of different models' capabilities across various legal NLP tasks.

Recently, specialized legal benchmarks designed explicitly for evaluating the performance of LLMs include datasets and tasks that specifically target legal language understanding and reasoning, providing a more nuanced and comprehensive assessment of LLMs' capabilities in legal contexts. LawBench (Fei et al., 2023) is a comprehensive benchmark for evaluating LLMs in the legal domain, assessing their abilities in legal knowledge memorization, understanding, and application across 20 diverse tasks. Extensive evaluations of 51 LLMs, including multilingual, Chinese-oriented, and legal-specific models, reveal GPT-4 as the top performer, indicating the need for further development to achieve more reliable legal-specific LLMs for related tasks. Table 9 summarizes the performance of various methods on JEC-QA dataset (Zhong et al., 2020), LEVEN dataset (Yao et al., 2022a), LawGPT dataset (Xiao-Song, 2024), and CAIL2018 dataset (Xiao et al., 2018). LEGALBENCH (Guha et al., 2023), another legal reasoning benchmark with 162 tasks across six legal reasoning types, created collaboratively by legal professionals. LEGALBENCH (Guha et al., 2023) aims to assess the legal reasoning capabilities of LLMs and facilitate cross-disciplinary dialogue by aligning LEGALBENCH (Guha et al., 2023) tasks with popular legal frameworks. An empirical evaluation of 20 LLMs is presented, showcasing LEGALBENCH (Guha et al., 2023)'s utility in guiding research on LLMs in the legal field. Complementing these, LAiW (Dai et al., 2023) focuses on the logic of legal practice, structuring its evaluation around the process of syllogism in legal logic. LAiW (Dai et al., 2023) divides LLMs capabilities into basic information retrieval, legal foundation inference, and complex legal application, across 14 tasks. The findings from LAiW (Dai et al., 2023) suggest that LLMs show proficiency in generating text for complex legal scenarios, but their performance in basic tasks is still unsatisfying. Additionally, while LLMs may exhibit robust performance, there remains a need to reinforce their ability of legal reasoning and logic.

## 5.4 LLM-based Methodologies for Legal Tasks and Challenges

This section discusses LLM-based approaches aimed at addressing significant challenges in Legal NLP. These challenges cover multiple aspects, including societal legal problems, legal prediction, document analysis, legal hallucinations, legal exams, and the need for robust LLM Agents.

**Societal Legal Challenges.** LLMs have emerged as powerful tools with the potential to address various societal challenges in daily life. In the area of legal applications, LLMs are being explored for their capabilities in areas such as tax preparation, online disputes, cryptocurrency cases, and copyright violations. For instance, the use of few-shot in-context learning could improve the performance of LLMs in tax-related tasks (Srinivas et al., 2023; Nay et al., 2024). Moreover, Llmediator (Westermann et al., 2023) highlights the role of LLMs in facilitating online dispute resolution, especially for individuals representing themselves in court, it generates dispute suggestions by detecting the inflammatory message and reformulating polite messages. Additionally, the exploration of LLMs in cryptocurrency security cases (Trozze et al., 2024) (Zhang et al., 2023k) demonstrates their utility in navigating intricate legal landscapes. Addressing copyright violations is another area where LLMs are making an impact (Karamolegkou et al., 2023).

**LLM Legal Prediction.** Legal prediction judgment is a crucial task of leveraging LLMs in the legal domain. Among the various techniques, Legal Prompt Engineering (LPE) stands out as a commonly used method for enhancing legal predictions. LPE (Trautmann et al., 2022) is a technique that enhances legal responses using key strategies like zero-shot learning, few-shot learning, the chain of reference (CoR), and RAG. Trautmann et al. (2022) show that zero-shot LPE is better compared to the baselines, but it still falls short compared to state-of-the-art supervised approaches. Kuppa et al. (2023) propose CoR, where legal questions are pre-prompted with legal frameworks to simplify the tasks into manageable steps, leading to a significant improvement in zero-shot performance by up to 12% in LLMs like GPT-3. Jiang & Yang (2023) introduce legal syllogism prompting (LoT), a simple method to teach LLMs for LJP, focusing on the basic components of legal syllogism: the major premise as law, the minor premise as fact, and the conclusion as judgment.

**LLM Document Analysis.** LLMs could also assist in legal document analysis, and be applied to case files and legal memos for content extraction. Contract management can be enhanced through automated drafting,

review, and risk assessment. LLMs facilitate the mining and analysis of legal cases in case and precedent studies. Steenhuis et al. (2023) outline three methods for automating court form completion: using GPT-3 in a generative AI approach to iteratively prompt user responses, employing a template-driven method with GPT-4-turbo for drafting questions for human review, and a hybrid approach. Choi (2023) discuss using LLMs for legal document analysis, assessing best practices, and exploring the advantages and limits of LLMs in empirical legal research. In a study comparing Supreme Court opinion classifications, GPT-4 matched human coder performance and outperformed older NLP classifiers, without gains from training or specialized prompting.

**Legal Hallucination Challenges.** With the advent of GPT-4, a surge in research has leveraged this advance to assist in legal decision-making, aiming to offer strategic legal advice and support to lawyers. However, this approach is not without its skeptics. A notable concern is the phenomenon of hallucinations (Zhang et al., 2023l), where the LLMs, despite uncertainties, may suggest decisions in cases when it has less confidence. This highlights a crucial area for further scrutiny, balancing the innovative potential of GPT-4 with the need for reliability and accuracy in sensitive legal contexts. Dahl et al. (2024) investigate the phenomenon of legal hallucinations in LLMs and explore the development of a typology for such hallucinations, the high prevalence of inaccuracies in responses from popular LLMs like GPT-3.5 and Llama 2, the models' inability to correct false legal assumptions, and their lack of awareness when generating incorrect legal information.

**LLM Agent Challenges.** Developing LLM agents is incredibly challenging due to their specialized design for various legal tasks like providing advice and drafting documents. Their role is crucial in improving legal workflows and efficiency, highlighting the difficulty in their development. For instance, Cheong et al. (2024) examine the implications of using LLMs as public-facing chatbots for providing professional advice, highlighting the ethical, legal, and practical challenges, particularly in the legal domain, and it suggests a case-based expert analysis approach to inform responsible AI design and usage in professional settings. Iu & Wong (2023) sense ChatGPT's potential as a substitute for litigation lawyers, focusing on its drafting abilities for legal documents such as demand letters and pleadings, noting its proficient legal drafting capabilities.

**Legal Exam Challenges.** Numerous attempts have been made to pass various judicial examinations using LLMs (Choi et al., 2021; Bommarito II & Katz, 2022; Martínez, 2024), GPT-4 passes the bar exam (Katz et al., 2024) but has a long way to go for LexGLUE (Chalkidis et al., 2021c) benchmark (Chalkidis, 2023). Yu et al. (2023a) further their application in legal reasoning by conducting experiments with the COLIEE2019 entailment task (Kano et al., 2019), which is based on the Japanese bar exam.

## 5.5 Future Prospects

**Building High-Quality Legal Datasets.** Considering the legal domain's intricate semantics and its requirements for precise statutes, obtaining high-quality legal datasets is often a particularly challenging task. More specifically, most existing legal datasets collected from the natural world are incomplete, sparse, and complicated. Its complexity and scholarly nature make it difficult for regular machine learning approaches to provide annotation, while manual annotation in the legal domain requires much higher demands and costs (e.g., legal training and expertise) than in general domains. For example, CUAD (Hendrycks et al., 2021b) was created with dozens of legal experts from the Atticus project (Contributors, 2024) and consists of over 13,000 annotations. In the future, building high-quality legal datasets may cover the following interesting directions:

- **Multi-Source Legal Data Integration for LLMs.** Real-world legal events often involve data from a multitude of different information sources such as court records, evidence documentation, and multimedia materials (Matoesian & Gilbert, 2018). These pieces of information often exhibit significant diversity, ranging from precise and accurate legal texts to trivial and irrelevant details, and even intentionally obfuscated or ambiguously confused testimonies. Integrating information from diverse sources requires advanced data integration techniques. This requires not only general data processing skills such as multi-modal data fusion but also an understanding of domain-specific nuances such as legal terminology and organizational structures. Additionally, real-world legal case handling often requires global information, especially for long-text legal data. LLeQA (Louis et al., 2024) has made a promising start in providing long-form answers to statutory law questions, paving

the way for further research on handling long-text data. This enhanced capability in processing long-form text will be important for addressing complex cases and offering comprehensive legal support. Future research focusing on the recognition of key patterns or legal symbols in long-form text may enhance the model's understanding of lengthy documents.

- **Legal Dataset Collection and Augmentation with LLMs.** Firstly, leveraging the capabilities of LLMs offers promising solutions to simplify the data collection process in the legal domain. More specifically, by harnessing the language processing abilities of LLMs, researchers can automate tasks traditionally requiring extensive knowledge and manual effort, such as legal document annotation and classification. Moreover, LLMs can bridge the knowledge gap between the NLP community and legal experts, enabling efficient extraction of relevant legal information from vast repositories of plain text. This can not only simplify data collection but also empower researchers to navigate complex legal documents with ease, facilitating the generation of high-quality datasets with minimal human work. Furthermore, another intriguing direction involves legal data augmentation in terms of varied formats, the dataset should not only comprise structured court documents but also leverage unstructured text from social media, news, and other sources for enrichment. For the few-shot low-frequency datasets within the LJP tasks, methods integrating data augmentation and feature augmentation are crucial (Wang et al., 2021). These augmentation methods effectively boost dataset diversity, thereby enhancing model performance and robustness.

**Developing A Comprehensive LLM-based Legal Assistance System.** As aforementioned, while LLMs have made progress in addressing several important legal tasks, their coverage over the legal domain is still far from comprehensive. Looking forward, our long-term objective can be developing practical and systematic legal assistance systems that benefit human life and bring positive social impact. Here, we focus on several specific scenarios where such systems can make a significant impact, including:

- **LLM-based Legal Advice.** As an ongoing and significant area of focus, providing legal advice presents a formidable challenge due to its reliance on legal domain knowledge, cultural context, and intricate logical reasoning. However, the prospects for leveraging LLMs in this domain are promising. LLMs have been trained on vast amounts of data, enabling them to embed cultural backgrounds and common sense into their understanding. Moreover, their capacity for comparing cases across jurisdictions and incorporating human knowledge into inference processes holds the potential for advancing legal reasoning capabilities. Despite the complexities involved, the integration of LLMs in legal reasoning stands to enhance efficiency and accuracy in legal decision-making processes. In this direction, some promising research topics would include: (1) Integrating advanced tools like knowledge graphs (Hogan et al., 2021; Huang et al., 2020) to enhance LLMs' legal domain knowledge and logical reasoning capabilities; (2) Considering the fact of data scarcity in existing legal data, incorporating user feedbacks regarding few-shot learning into LLM-based legal advice is vital. These feedbacks may be collected from users with varying levels of legal expertise, and they can also be used to prevent the dissemination of unethical or inaccurate advice. (3) As LQA in legal scenarios requires complex logical reasoning, it naturally leads to future research directions of enhancing the parameters scale for legal-specific LLMs. On the other hand, to improve the efficiency of real-time LQA, research on legal LLM compression would be important in real-world applications.

- **LLM-based Legal Explanation and Analysis.** Existing LLM-based methods often operate like black boxes, thus legal case explanation and analysis represent another critical task in the legal domain. This includes both the examination of real-world court cases and the decisions made by LLMs. In terms of LLM-generated decisions, an intriguing path of research involves developing mechanisms for self-explanation akin to CoT (Wei et al., 2022) approaches. For real-world legal cases, LLMs can offer context-specific explanations tied to real-world scenarios when analyzing them. Moreover, LLMs have the potential to provide various types of human-understandable explanations, spanning general legal regulations, example-specific discussions, and comparisons between analogous cases—potentially transcending state, time, and country boundaries. Such multifaceted explanations can enhance the trustworthiness and transparency in the legal domain, mitigating unfairness and

ethical issues like gender bias in legal systems (Sevim et al., 2023), reducing legal hallucination significantly (Dahl et al., 2024).

- **Social Impact of LLM-based Legal System.** Investigating the social impact of LLMs on the legal domain also includes many interesting directions: (1) Applying LLMs to democratize legal education and advice, benefiting individuals who have difficulties in visiting a human lawyer due to the lack of professional knowledge or economic resources. This democratization can empower marginalized communities by granting them access to crucial legal information and guidance; (2) The development of LLMs will also accelerate the evolution of privatized and personalized legal LLMs, leading to increasing competition in the legal domain and the creation of more satisfying products for customers (Cui et al., 2023a); (3) Leveraging LLMs to drive future developments in law through enhanced legal analysis. By facilitating deeper insights into legal texts and precedents, LLMs can contribute to more informed law updates and academic research endeavors; (4) Addressing ethical issues with LLMs in the legal system. Through rigorous analysis and scrutiny, LLMs can help identify and rectify instances of injustice and discrimination against certain demographic groups in legal decision-making processes.

# 6 Ethics

Despite recent breakthroughs in LLMs, concerns regarding their ethics and trust have been raised for their real-world use (Kaddour et al., 2023; Ray, 2023). Especially, when applied in high-stakes domains such as FHL, these ethical concerns become particularly critical. In a broader sense, the ethics of AI technologies in different domains have been widely discussed in the last few decades (Jobin et al., 2019; Leslie, 2019). Despite these large number of existing discussions, numerous ethical concepts are proposed from diverse disciplines and perspectives with complicated objectives, leading to challenges to constructing a consistent and well-organized ethical framework. Fortunately, these various ethical considerations often originate from similar high-level principles. In this section, we first introduce several general ethical principles and related considerations for LLM applications, and also showcase examples of these ethics in domain-specific contexts. We will describe the basic definitions of these ethical issues and summarize the existing investigations for testing or addressing them in section 6.1, then we discuss future directions in section 6.2.

## 6.1 Ethical Principles and Considerations

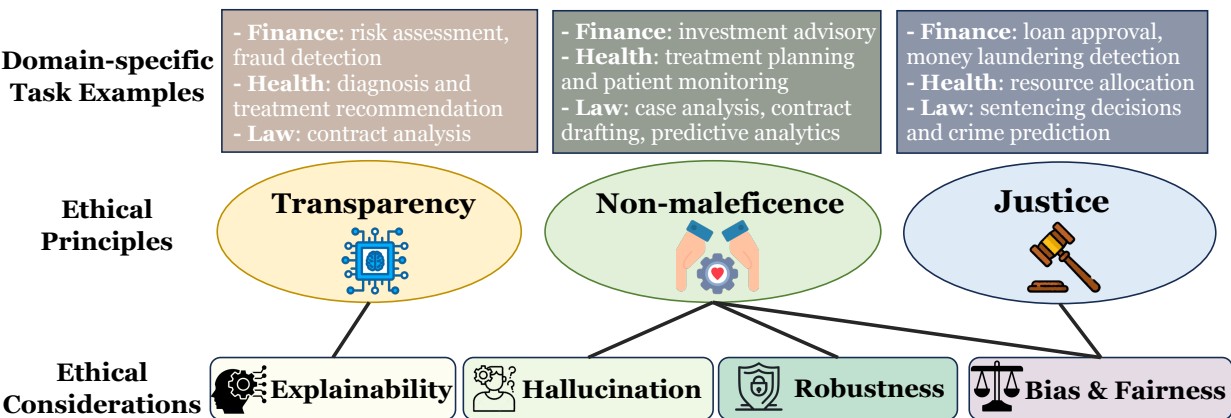

Figure 6: Important ethical principles, considerations, and domain-specific examples in finance, healthcare, and law.

In the discourse on AI ethics, there are several general principles that find broad adoption. Guided by these principles, a multitude of nuanced definitions are elaborated across various domains. For example, an investigation (Jobin et al., 2019) on 84 AI ethical documents summarizes 11 frequent ethical principles and

guidelines, including transparency, justice, responsibility, non-maleficence, privacy, beneficence, autonomy, trust, sustainability, dignity, and solidarity. Here, combining the most urgent ethical concerns regarding LLMs and the varying focuses across FHL domains, we mainly highlight three **ethical principles** (*Transparency, Justice, Non-maleficence*) and several most prevalent **ethical considerations** (*Explainability, Bias&Fairness, Robustness, Hallucination*) that are associated with these principles. Figure 6 shows the connection between these ethical principles and considerations, as well as some example tasks that prioritize these ethical principles in different domains.

### 6.1.1 Ethical Principles

**Transparency.** Transparency refers to "explaining and understanding" the systems, including different stages such as data usage and model behavior. Transparency is the most frequently mentioned AI ethical principle based on the investigation in Jobin et al. (2019). Many concepts are related to transparency, such as explainability, interpretability, communication, and accountability. Transparency is particularly critical when LLM assists in complicated, expertise-intensive, and high-risk applications. In finance, institutions have started to utilize LLMs for tasks such as risk assessment, fraud detection, and automated trading strategies; In healthcare, LLMs are increasingly employed in clinical decision support, such as disease diagnosis and treatment recommendation; In law, LLMs have been used for contract review and analysis. In these examples of applications, transparency is crucial to promote understanding of how LLMs make final decisions and thereby assess their potential risks or issues.

**Justice.** Justice encompasses a spectrum of meanings, commonly associated with "fairness, equity, inclusion, diversity, non-bias, and non-discrimination". Hence, justice holds particular significance when LLMs are utilized in contexts involving individuals from diverse demographic or societal backgrounds. In studies of law, justice is often widely referenced as a core legal principle. For instance, when using LLMs to assist in sentencing decisions or crime prediction, justice (e.g., anti-discrimination against race, economic/political status, and crime history) is crucial. In finance, applications such as loan approval, money laundering detection, and consumer rights protection have high demands for fairness and equity. In healthcare, fair resource allocation and treatment recommendations without inequalities and discrimination are of great importance when LLMs are applied.

**Non-maleficence.** Non-maleficence generally means "do no harm". Here, harms can exist in a variety of forms such as incorrect, toxic, outdated, biased, and privacy-violating information. As LLMs are often trained on vast corpora with unknown quality, the removal of such harmful information is imperative in practical applications. In finance, non-maleficence is important because it emphasizes the responsibility of financial institutions, professionals, and regulators to prevent harm to investors, consumers, and the broader financial system, avoiding potential financial losses or harm to individuals or society. In healthcare, non-maleficence plays a key role in maintaining patient safety, trust, and confidentiality, especially in tasks like treatment planning and patient monitoring. In law, non-maleficence is crucial to prevent wrong actions, negligence, and violations of legal rights stemming from reliance on obsolete or inaccurate legal provisions. Especially, due to the dynamic nature of legal systems, updated law may inadvertently leave behind outdated and harmful information, necessitating careful approaches to ensure that legal practices and interpretations remain aligned with current statutes and regulations.

### 6.1.2 Ethical Considerations

**Explainability.** Explainability means the capacity to elucidate the model behavior in a human-understandable way (e.g., showing the importance of input data or model component for model output, and estimating the model behavior in interventional or counterfactual cases). AI explainability has been a longstanding concern (Saeed & Omlin, 2023; Došilović et al., 2018), as many AI models inherently function as black boxes, lacking transparency and interoperability. Especially, explanation in LLMs is often even more challenging than most traditional AI techniques due to the extensive scale of training data and the large size of the model. Despite the challenges, from another aspect, the unique ability of LLMs to comprehend and generate natural language empowers them to elucidate their own decision-making processes. Recent investigations (Zhao et al., 2024; Singh et al., 2024) have summarized existing explanation approaches for LLMs in both traditional fine-tuning paradigm (with approaches such as feature-based explanation (Ribeiro et al., 2016; Lundberg & Lee, 2017) or

example-based explanation (Koh & Liang, 2017; Verma et al., 2020)), and recent prompting-based paradigm (with approaches such as in-context learning explanation (Li et al., 2023j) and CoT prompting explanation (Wu et al., 2023e)).

**Bias & Fairness.** Bias and fairness broadly include various ethical terms, such as social stereotypes or discrimination against certain demographic groups related to sensitive features (e.g., race, gender, or disability) (Gallegos et al., 2023; Ghosh & Caliskan, 2023) and monolingual bias (Talat et al., 2022) for the languages different from training data. Uncensored natural language often contains numerous biases, and the culture, language, and demographic information included in training corpora are often highly imbalanced, which are the main causes for unfair language models. Besides, improper model selection and learning paradigms can also lead to biased outcomes. Existing work (Gallegos et al., 2023; Kotek et al., 2023; Zhuo et al., 2023; Ghosh & Caliskan, 2023; McGee, 2023; Motoki et al., 2023) has discussed and evaluated LLMs in terms of bias in different cases, showing that LLMs have a certain ability to resist social discrimination in open-ended dialogue, but still often exhibit varying forms of bias. With such issues, many efforts have been made to mitigate bias in LLMs (Li et al., 2023e; Ferrara, 2023; Gallegos et al., 2023), covering both perspectives from data-related bias or model-related bias. Current debiasing methods mainly include (a) *Pre-processing* approaches which mitigate bias in LLM by changing the model training data, such as data augmentation and generation (Xie & Lukasiewicz, 2023; Stahl et al., 2022), as well as data calibration (Ngo et al., 2021; Thakur et al., 2023; Amrhein et al., 2023) and reweighting (Han et al., 2021; Orgad & Belinkov, 2023); (b) *In-processing* approaches that debias by changing LLM models. Multiple technologies such as contrastive learning (He et al., 2022; Li et al., 2023f), model retraining (Qian et al., 2022), and alignment (Guo et al., 2022; Ahn & Oh, 2021) have been adopted in these studies; (c) *Post-processing* methods that mitigate bias from model outputs (Liang et al., 2020; Lauscher et al., 2021; Dhingra et al., 2023).

**Robustness.** Although its definition varies in different contexts, robustness generally denotes a model's capacity to sustain its performance even for input that deviates from the training data. The deviation can be triggered by different factors such as cross-domain distributions and adversarial attacks. Models lacking robustness often result in a cascade of adverse consequences, e.g., privacy leakage (Carlini et al., 2021), model vulnerability (Michel et al., 2022), and generalization issues (Yuan et al., 2024). Various attacks (Zou et al., 2023; Lapid et al., 2023; Liu et al., 2023e; Wei et al., 2024; Shen et al., 2023b; Zhuo et al., 2023; Shi et al., 2024) for LLMs are emerging continuously. Some of them inherit commonly used attacking strategies in traditional domains such as computer vision (Szegedy et al., 2013; Biggio et al., 2013). Others explore "jailbreaks" (Liu et al., 2023e; Wei et al., 2024; Deng et al., 2023a), aiming to strategically craft prompts (with human effort or automatic generation) to result in outputs deviating from the purpose of aligned LLMs. Further studies also focus on the universality and transferability of attacks (Zou et al., 2023; Lapid et al., 2023). These studies found that numerous vulnerabilities and deficiencies in LLMs persist, prompting severe societal and ethical issues. Simultaneously, there has been a surge in literature (Yuan et al., 2024; Altinisik et al., 2022; Stolfo et al., 2022; Moradi & Samwald, 2021; Shi et al., 2024; Ye et al., 2023b; Mozes et al., 2023; Wang et al., 2023b; Schwinn et al., 2023; Jain et al., 2023; Kumar et al., 2023) dedicated to researching and evaluating the LLM robustness. Previous work (Wang et al., 2023e) categorizes existing methods against jailbreak attacks on LLMs into two directions: internal safety training (Ganguli et al., 2022; Touvron et al., 2023b) (i.e., further train the LLM model with adversarial examples to better identify attacks) and external safeguards (Jain et al., 2023; Markov et al., 2023) (i.e., incorporate external models or filters to replace harmful queries with predefined warnings). SELF-GUARD (Wang et al., 2023e) combines these two types of safety methods. It is also worth mentioning that self-evaluation for LLM outputs (Helbling et al., 2023; Li et al., 2023h) has become an emerging trend in defense strategies.

**Hallucination.** Hallucination has been a ubiquitous issue in NLP, which refers to the generation of incorrect, nonsensical, or misleading information. Especially, hallucination in LLMs faces unique challenges due to its difference from traditional language models (Ji et al., 2023). The authenticity and precision of pursuits in FHL domains inevitably underscore the pressing concern of hallucination (Alkaissi & McFarlane, 2023). Mainstream work (Ji et al., 2023; Maynez et al., 2020; Kaddour et al., 2023) categorizes hallucination into two types: (1) *Intrinsic hallucination*: the generated output conflicts with the source content (e.g., the prompt); and (2) *Extrinsic hallucination*: the correctness of the generated output cannot be verified based on the source content. Although extrinsic hallucination is not always incorrect, and sometimes can even provide useful

background information (Maynez et al., 2020), we should still handle any unverified information cautiously. Current work (Min et al., 2023b;b; Ren et al., 2023) has identified and evaluated hallucination in different ways, including those based on external verified knowledge such as Wikipedia (e.g., Kola (Yu et al., 2023b), FActScore (Min et al., 2023b), FactualityPrompts (Lee et al., 2022)), as well as those based on probabilistic metrics such as uncertainty of LLM generation (Manakul et al., 2023; Varshney et al., 2023). Many factors can result in hallucination, such as biases inside data, outdated corpora, prompt strategy, and intrinsic model limitations. Correspondingly, as discussed in Ji et al. (2023), existing approaches for LLM hallucination limitation cover different branches including data-centric and model-centric. Data-centric methods eliminate hallucination by improving the data quality in different stages (Zhang et al., 2023f; Penedo et al., 2023; Es et al., 2023). Model-centric approaches focus on the model design and their training or tuning procedure. Representative methods in this line include reinforcement learning with human feedback (RLHF) (Ouyang et al., 2022), model editing (Daheim et al., 2023), and decoding strategies (Dziri et al., 2021; Tian et al., 2019).

### 6.1.3 Domain-specific Ethics

On top of general ethical principles and considerations, in the specific context of different domains, the definitions of ethics display their distinct focus and subtle differences. Here, we introduce domain-specific investigations of ethics in FHL domains, respectively.

**Finance.** Many ethical guidelines (Attard-Frost et al., 2023; Svetlova, 2022; Kurshan et al., 2021; Farina et al., 2024) for AI practice in the finance sector have been published in recent years. In Attard-Frost et al. (2023), based on the general Fairness, Accountability, Sustainability, and Transparency (FAST) AI ethical principles in the public sector proposed by Leslie (2019), a series of business-oriented ethical themes (e.g., market fairness, bias & diversity in professional practice, and business model transparency) are organized under each principle. When using LLMs for finance, a few studies have begun to discuss the ethics of LLMs such as ChatGPT (Khan & Umer, 2024) and BloombergGPT (Wu et al., 2023d). Exploratory efforts for addressing ethical issues of LLMs in finance, such as hallucination (Kang & Liu, 2024; Roychowdhury et al., 2023) and financial crime (Ji et al., 2024), have laid a promising groundwork for further investigation.

**Healthcare.** Ethics in healthcare has long garnered significant attention (Pressman et al., 2024; Beauchamp & Childress, 2001) due to potentially severe and irreversible consequences, notably the loss of human life. As a result, a set of widely adopted ethical principles (Autonomy, Beneficence, Non-maleficence, and Justice) (Beauchamp & Childress, 2001) has been established in clinical and medical practice. Apart from the aforementioned non-maleficence and justice, autonomy in health centers on an individual's right to make informed medical decisions, and beneficence in health focuses on "doing good" to promote patient well-being. Recent discussions (Li et al., 2023b; Karabacak & Margetis, 2023; Minssen et al., 2023; Yu et al., 2023c; Thirunavukarasu et al., 2023; Haltaufderheide & Ranisch, 2024; Ullah et al., 2024) for the ethics of LLMs in the health & medicine sector have reached the consensus that existing LLMs still have a substantial gap to bridge in order to meet ideal ethical standards. This situation leads to the development of more nuanced ethical considerations across various healthcare scenarios. For instance, a recent review (Haltaufderheide & Ranisch, 2024) summarized LLM ethics in four key clinical themes, including clinical applications, patient support, health professionals, and public health. Other discussions about ethics in specific healthcare contexts such as surgery (Pressman et al., 2024) and mental health (Cabrera et al., 2023) also provide valuable insight into LLM applications in real-world health systems.

**Law.** In the domain of law, numerous deliberations (Cranston, 1995; Yamane, 2020; Wright, 2020; Nunez, 2017) has taken place about legal ethics for AI. The recent progress of LLMs brings new challenges and discussions about ethics in the legal domain, stimulating the refinement of existing legal ethics and the development of more feasible evaluation standards. Among these works, Zhang et al. (2024a) design a multi-level ethical evaluation framework and evaluates mainstream LLMs under the framework. This evaluation framework covers three aspects with increasing level of ethical proficiency: legal instruction following (i.e., the ability of LLMs to address user needs based on given instructions), legal knowledge (i.e., the ability of LLMs to distinguish the legal/nonlegal elements), and legal robustness (i.e., the consistency of LLM responses to identical questions presented in varying formats and contexts). Another recent work (Cheong et al., 2024) collects opinions from 20 legal experts, revealing detailed policy considerations for LLM employment in the

professional legal domain. Moreover, a few exploratory works in more specific tasks such as profiling legal hallucinations also start to attract people's attention (Dahl et al., 2024). These studies set the stage for more comprehensive ethics regulations in future "LLM + Law" applications.

## 6.2 Future Prospects

### 6.2.1 Methodological Directions

Looking ahead, the future prospects for LLM in FHL domains are at an exciting moment. Here, we outline several promising methodological directions that can be applied to address various ethical concerns:

- **Dataset censorship.** Meticulous dataset censorship is vital, involving a thorough examination and elimination of improper content from the training data. This step ensures that the model is shielded from potentially harmful information, minimizing the risk of encoding unwanted patterns. For example, removing biased, private, or incorrect information in FHL scenarios can promote fairness, privacy, and reduction of hallucinations. Considering the specialized knowledge required in FHL domains, developing high-quality data censorship mechanisms presents significant challenges.

- **Human and domain knowledge for ethics.** The integration of humans in the AI loop is essential. Human reviewers contribute nuanced perspectives, provide domain knowledge, identify ethical issues, and guide the model's learning process by refining its responses. Human-in-the-loop systems allow for ongoing monitoring and adjustments to address emerging ethical problems. For example, as discussed in Section 5, involving legal advice from human expert would not only improve legal reasoning but also address complicated concerns in ethical and moral dilemmas.

- **Theoretical bounds.** Establishing theoretical bounds on the model's behavior is important. The development of clear theoretical frameworks and ethical guidelines helps delineate the limits of the model's decision-making, preventing it from generating potentially harmful or biased outputs. Through the implementation of these measures, we can elevate the ethical standards of LLMs, fostering responsible AI development. For example, for the robustness issue we discussed in Section 6.1.2, certifiable approaches for LLMs are crucial to theoretically guarantee model safety against a range of adversarial attacks.

- **Causality-involved analysis.** It is essential to delve into the underlying causes and mechanisms behind the generation of LLM outputs. For example, concerning the explainability and fairness issues discussed in Section 6.1.2, incorporating causality can help elucidate the causes behind model behavior and eliminate potential biases against underrepresented groups. By understanding and addressing the causal relationships within the data and the model, we can develop effective strategies to improve both the transparency and equity of the LLM outputs.

### 6.2.2 Further Ethical Concerns in FHL

Here, we highlight several urgent and critical concerns for LLM + FHL areas, which hold substantial potential for future development.

**Safety and privacy.** Employing LLMs in FHL sectors introduces substantial safety and privacy issues, given the sensitive nature of the information handled in these areas. Existing studies (ThankGod Chinonso, 2023) have reported LLM issues in privacy and safety, coming from both model-inherent vulnerabilities (e.g., data extraction, data poisoning) and other vulnerabilities (e.g., prompt injection), as we also introduced in Section 6.1. Traditional privacy and safety strategies often fail on LLMs due to the large scale of LLM parameters and high computation cost. Considering this, we list the following lines for future research: (1) **Clear guidelines and user education:** Provide clear, understandable guidelines that inform FHL users about the risks associated with the provision of personal information. Regularly remind and educate users on the importance of being cautious with the amount of personal information they share. Emphasize the

use of minimal and relevant data inputs when interacting with LLMs to reduce privacy risks. (2) **Vigilant third-party management:** Implement a rigorous screening process for all third-party partners who may have access to the data used by LLMs. Establish strict data handling and confidentiality agreements with third-party partners to prevent data leakage. These agreements should enforce adherence to privacy standards and include penalties for non-compliance. (3) **Robust defense strategies for LLMs in FHL:** Develop a FHL domain-specific security framework for LLMs to protect against unauthorized access and data breaches. Regularly evaluate and update defense strategies to address emerging security threats, and establish a robust incident response plan to handle potential security breach.

**Explainability.** We have discussed the general considerations and methods for explanability in Section 6.1.2. In many settings, LLM applications in FHL can inherit or extend general LLM explanation methods. On top of that, in special FHL context that general explanation methods cannot tackle, some promising directions may include: (1) **Domain-specific customization:** Tailor explanations to the specific needs and comprehension levels of different stakeholders in FHL sectors, recognizing that financial experts, healthcare providers, and legal professionals may require different types of information to effectively trust and use AI outputs. (2) **Feedback loops for continuous improvement:** As FHL domains often involve long-term or multi-stage processes, we need to establish mechanisms for users to provide feedback on explanations, which can be used to continually refine explanation methodologies.

**Potential exacerbation of existing inequalities.** As the bias issue we discussed in Section 6.1, LLM applications in FHL areas may suffer from exacerbation of inequalities. We suggest the following directions for future studies: (1) **Definition of FHL domain-specific bias and fairness notions:** On top of general bias notions such as group fairness and individual fairness, it is crucial to first establish clear definitions of bias and fairness specifically in the FHL domains. These definitions should reflect the unique characteristics and requirements of each sector that have not been covered in existing general bias problems. (2) **Generative FHL data to eliminate bias:** One promising direction to combat bias is the generation of synthetic data that reflects the diversity and complexity of real-world scenarios in each FHL domain. By enriching training datasets with generative techniques, researchers can better model minority cases and underrepresented groups. (3) **Legacy support and impact assessments:** Integrate bias and fairness impact assessments as part of the compliance process for FHL-focused LLMs. Such assessments should be regularly updated and publicly accessible to maintain trust and accountability.

**Job displacement.** Widespread adoption of LLMs could lead to significant workforce changes and potential job losses. But it is important to consider that LLMs bring efficiency improvement for routine and entry-level tasks; meanwhile, they also present opportunities for workforce enhancement and the creation of new job roles. For instance, while routine tasks may be automated, there will be an increased demand for skilled professionals to manage, interpret, and oversee these technologies. This shift could lead to the emergence of new specialties and roles that are currently unforeseen, contributing positively to job creation in the economy. To address the potential for job changes, we suggest several potential proactive actions: (1) **Upskilling and reskilling programs:** Implement training programs to help current employees adapt to new roles that require managing or working alongside LLM technologies. (2) **Policy development:** Encourage the creation of policies that support workforce transitions. This includes social safety nets and incentives for businesses to retrain rather than replace their workforce. Foster collaboration between policymakers, educational institutions, industry leaders, and workers' representatives to ensure a holistic approach to the challenges and opportunities posed by LLMs.

**Accountability and liability.** We suggest the following points to be considered for all stakeholders: (1) **Clear attribution of responsibility:** Define clear guidelines for attributing responsibility among all stakeholders involved in the development and deployment of LLMs, including developers, deployers, and end-users to help establish a chain of accountability. (2) **Scenario-based guidance:** Explore various scenarios where LLM outputs could lead to errors or harm and propose specific guidance on how accountability and liability should be managed in each case. This approach can help clearly prepare stakeholders for potential issues and accountability. (3) **Stakeholder collaboration:** Encourage collaboration among legal experts, ethicists, technologists, and policymakers to develop a comprehensive framework that addresses accountability and liability. This multidisciplinary approach is essential for creating robust solutions that are practical and enforceable.

**Widening technology gaps.** Smaller organizations and non-profits may face barriers to accessing advanced LLMs, potentially exacerbating disparities in service quality. Potential directions to address this issue may include: (1) **Open-source initiatives:** Promote the development and use of open-source LLM tools that could reduce costs and increase accessibility for all organizations. (2) **Public-private partnerships:** Facilitate partnerships between government bodies, academic institutions, and private sector leaders to provide the necessary resources and training to under-resourced groups. (3) **Subsidized pricing models:** Encourage commercial LLM developers to offer tiered pricing or subsidies for non-profits. (4) **Policy recommendations:** Propose specific policy interventions that could support wider access to these technologies. This might include government-funded programs to aid smaller entities in adopting LLMs or incentives to support non-profit and smaller organizations. (5) **Awareness and training programs:** Suggest the implementation of targeted programs to raise awareness and improve the AI literacy of smaller organizations and non-profits, enabling them to effectively use and benefit from LLM technologies.

**Risk assessment of unintended consequences.** LLM adoption could have unforeseen effects on FHL applications. We suggest the following strategies to handle unintended consequences: (1) **Risk assessment and management:** Develop systematic models for risk assessment that anticipate potential negative impacts of LLMs on FHL tasks. This includes predicting volatility, assessing the risk of bias, and identifying points of failure. (2) **Continuous monitoring:** Implement continuous monitoring systems that can dynamically assess and manage the risks associated with operational LLMs. These systems would use real-time data to update risk profiles and suggest necessary adjustments to deployment strategies. (3) **Consequence explanability:** Develop and enforce standards for explainability that require LLMs to provide understandable and detailed justifications for their outputs and potential outcomes at each step, so that developers can ensure that systems are easier to audit and improve over time.

**Public trust and acceptance.** Building public trust in AI systems is essential for their successful deployment. Potential future strategies to build and maintain public trust in LLMs may include: (1) **Transparency initiatives:** Make the algorithms' decision-making processes more accessible and understandable to the general public, as well as disclose any limitations and uncertainties associated with AI predictions. (2) **Educational outreach:** Suggest strategies for educational outreach to improve public understanding of AI. This could include partnerships with educational institutions to integrate AI literacy into curriculums or public workshops and seminars that explain how LLMs are used in specific domains. (3) **Stakeholder involvement:** Recommend involving a broad range of stakeholders in the development and implementation of LLM policies. This should include policymakers, community leaders, consumer advocates, LLM developers, and other representatives from the society to ensure that multiple perspectives are considered and that AI deployments are aligned with public values and needs.

# 7 Conclusion

The exploration of LLMs across diverse fields illuminates the vast potential and inherent challenges of integrating advanced AI tools into various real-world applications. This survey focuses on three critical societal domains: finance, healthcare & medicine, and law, underscoring the transformative impact of LLMs in enhancing research methodologies and accelerating the pace of knowledge discovery and decision-making in these domains. Through detailed examination across disciplines, we highlight significant advancements achieved by leveraging LLMs in these domains, foreseeing a promising future full of breakthroughs and opportunities.

However, the integration of LLMs also brings to light challenges and ethical considerations. Concerns such as explainability, bias & fairness, robustness, and hallucination necessitate ongoing scrutiny and development of mitigation strategies. Furthermore, the interdisciplinary nature of LLM applications calls for collaborative efforts among AI researchers, domain experts, and policymakers to navigate the ethical landscape and harness the full potential of LLMs responsibly. As LLMs continue to evolve and find broader utility, it becomes increasingly imperative to address these challenges systematically and proactively.

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
