# OpenReview forum: "A Survey on Large Language Models for Critical Societal Domains: Finance, Healthcare, and Law"
_TMLR — Accepted by TMLR_

### Review · Reviewer_3Mk6 · 2024-08-20

**Summary Of Contributions:**

This paper is a comprehensive survey on the application of Large Language Models (LLMs) in critical societal domains such as finance, healthcare, and law.

1. The survey delves into the methodologies utilized by LLMs, offering insights into how these models are trained and applied within the specified domains. It highlights the instrumental role of LLMs in enhancing diagnostic and treatment methodologies in healthcare, innovating financial analytics, and refining legal interpretation and compliance strategies.

2. The paper critically examines the challenges faced by LLMs, including the need for professional expertise, handling of highly confidential data, interpretation of extensive multimodal documents, and adherence to high legal risk and strict regulations.  It points out existing ethical concerns and the necessity for transparent, fair, and robust AI systems that respect regulatory norms within these high-stakes sectors.

3. The survey underscores the imperative for interdisciplinary cooperation, methodological advancements, and ethical vigilance to maximize the benefits of LLMs while mitigating their risks. By presenting a thorough review of current literature and practical applications, the paper showcases the transformative impact of LLMs in finance, healthcare, and law.

4. It includes performance comparisons and analysis for various LLMs, offering insights and guidance for future research. The paper outlines potential future research directions, emphasizing the need for more sophisticated tasks, incorporation of multimodal documents, and fostering interdisciplinary collaboration and learning.

**Audience:**

Yes

**Broader Impact Concerns:**

Here are some key areas of concern:

1. Given the sensitive nature of data in finance, healthcare, and law, there is an urgent need to ensure robust data protection measures are in place to prevent breaches and maintain confidentiality.

2. LLMs may inadvertently perpetuate or amplify biases present in their training data, leading to unfair outcomes. It is crucial to address these concerns and ensure models are audited for bias.

3. The 'black box' nature of LLMs can make it difficult to understand their decision-making processes, which is problematic in domains where explainability is crucial for trust and regulatory compliance.

**Claims And Evidence:**

Yes

**Requested Changes:**

refer to weakness

**Strengths And Weaknesses:**

### Strongth:

The survey provides an extensive overview of LLMs across three critical domains, offering a multidisciplinary perspective. It includes detailed exploration of methodologies, applications, challenges, and opportunities, which is beneficial for both researchers and practitioners. The emphasis on ethics is particularly strong, with a dedicated section that discusses ethical principles and considerations in depth.


### Weakness:

1. While the survey is comprehensive, it might benefit from more detailed descriptions of the specific methodologies used in each domain. The inclusion of more case studies or examples demonstrating the application of LLMs in real-world scenarios could strengthen the practical implications of the research. Some readers may find the lack of technical depth in certain sections a limitation, particularly regarding the inner workings of LLMs and the specifics of their training processes.

2. Given the emphasis on ethics, a more explicit discussion on data privacy and security measures within the context of LLMs could be beneficial. The survey could delve deeper into the issues of cultural and demographic bias in LLMs, especially since these models are increasingly being used in diverse global contexts.

3. While the paper touches on regulatory compliance, a more detailed discussion on how LLMs can be integrated within existing legal and financial regulatory frameworks would be valuable.

4. For the research to be fully robust, the authors may consider providing more information on how their findings can be reproduced, including data sources and experimental setups.

---

> ### Author Response · Authors · 2024-09-06
> **Response to Reviewer 3Mk6**
>
> We appreciate the reviewer's valuable feedback and constructive suggestions.
>
> # Q: Need the inclusion of more case studies or examples demonstrating the application of LLMs in real-world scenarios
> In finance, healthcare, and law, domain-specific LLMs do not typically introduce new architectural innovations but focus on adapting general-domain models by utilizing domain-specific data and customized training approaches. These adaptations generally involve a mix of pre-training, fine-tuning, and instruction-based fine-tuning strategies, enabling the models to effectively cater to the specific needs of each domain. The detailed method discussion and examples in finance, health, and law are in Section 3.2, 4.2, and 5.2, respectively.
>
> # Q: More discussion in data privacy and security
> We have discussed model security in Section 6.1.2. Beyond that, we further added more in-depth discussion about privacy and security in Section 6.2. In general, LLM issues in privacy and safety come from both model-inherent vulnerabilities (e.g., data extraction, data poisoning) and other vulnerabilities (e.g., prompt injection). Traditional privacy and safety strategies often fail on LLMs due to the large scale of LLM parameters and high computation cost. Considering this, we list the following lines for future research:
>  - **Clear guidelines and user education**: Provide clear, understandable guidelines that inform FHL users about the risks associated with the provision of personal information. Regularly remind and educate users on the importance of being cautious with the amount of personal information they share. Emphasize the use of minimal and relevant data inputs when interacting with LLMs to reduce privacy risks.
> - **Vigilant third-party management**: Implement a rigorous screening process for all third-party partners who may have access to the data used by LLMs. Establish strict data handling and confidentiality agreements with third-party partners to prevent data leakage. These agreements should enforce adherence to privacy standards and include penalties for non-compliance.
> - **Robust defense strategies for LLMs in FHL**: Develop a FHL domain-specific security framework for LLMs to protect against unauthorized access and data breaches. Regularly evaluate and update defense strategies to address emerging security threats, and establish a robust incident response plan to handle potential security breach.
>
> # Q: How LLMs can be integrated within existing legal and financial regulatory frameworks
> A: We propose the following strategies for integrating LLMs into legal and financial regulatory frameworks:
> - **Compliance Automation**: LLMs can automate the analysis of legal texts and financial regulations, helping organizations identify relevant requirements and risks to ensure compliance.
> - **Real-Time Monitoring and Risk Assessment**: LLMs can monitor transactions, contracts, and market data, identifying potential violations or risks in real-time, and enhancing proactive risk management.
> - **Contract Drafting and Review**: LLMs assist in drafting and reviewing legal documents, ensuring they align with current regulatory frameworks and reducing human error.
> - **Cross-Jurisdictional Compliance**: LLMs help organizations navigate international regulations by providing guidance o=n compliance with multiple legal systems.
> - **Enhanced Reporting**: LLMs can streamline the creation of regulatory reports, improving accuracy and transparency across sectors.
>
> # Q: How their findings can be reproduced, including data sources and experimental setups
> A: All the mentioned findings regarding data sources, code, experiments, and literature are well cited with corresponding literature, where reproducing instructions can be found.
>
> # Q: Broader impact concern
> A: We have added a subsection "Further Ethical Concerns in FHL" for further discussion in Section 6.2 to address the broader impact concerns (including safety & privacy, explainability, bias, etc.)

---

### Review · Reviewer_Tyj3 · 2024-08-21

**Summary Of Contributions:**

This survey paper provides a comprehensive overview of the applications of large language models (LLMs) in three critical societal domains: finance, healthcare, and law. The authors aim to explore the methodologies, applications, challenges, ethics, and future opportunities of LLMs within these domains. The paper is structured into sections covering each domain, discussing relevant tasks, datasets, domain-specific LLMs, evaluation frameworks, and LLM-based methodologies. It also includes a section on ethical considerations across these domains.

This survey paper explores the applications, methodologies, challenges, and future prospects of large language models (LLMs) in three critical societal domains: finance, healthcare, and law. The key contributions of this work include:

1. A thorough overview of existing NLP tasks, datasets, and LLM-based approaches in each domain, providing a valuable resource for researchers and practitioners.

2. Analysis of domain-specific LLMs and evaluation of their performance compared to general-purpose LLMs on various tasks.

3. Discussion of LLM-based methodologies developed to address key challenges in each domain, such as data scarcity, multimodal understanding, and quantitative reasoning.

4. Examination of ethical considerations and challenges in applying LLMs to these high-stakes domains.

5. Identification of future research directions and opportunities for advancing LLM applications in finance, healthcare, and law.

The paper seeks to make a contribution by synthesizing a large body of recent studies on LLMs across these three domains, which are characterized by their reliance on professional expertise, confidential data, and strict regulations. By bringing together insights from these related fields, the authors wish to highlight some of the common challenges and opportunities for cross-pollination of ideas.

In my opinion, the survey is timely given the rapid pace of LLM development and growing interest in domain-specific applications. It serves as a valuable reference for researchers looking to understand the current state-of-the-art and identify promising avenues for future work. The authors’ commitment to maintaining an updated reading list is also commendable and might enhance the long-term value of this contribution to the research community, though I must admit that I am not sure whether it is truly feasible to maintain such a demanding task in the long term.

**Note:** I am currently recommending a light rejection only because I believe the paper might benefit tremendously from going through another round of reviews to iron out some major issues outlined in the following sections.

**Audience:**

Yes

**Broader Impact Concerns:**

This paper touches on several critical societal domains where the deployment of LLMs could have significant impacts, both positive and negative. Some key broader impact concerns that should be more thoroughly explored include:

1. **_Potential exacerbation of existing inequalities:_** LLMs trained on historical data may perpetuate or amplify biases related to race, gender, socioeconomic status, etc. in financial, healthcare, and legal decisions. More discussion is needed on approaches to identify and mitigate these risks.

2. **_Job displacement:_**  Widespread adoption of LLMs in these domains could lead to significant workforce changes and potential job losses, particularly for entry-level or routine tasks. The societal implications of this shift deserve more attention.

3. **_Over-reliance on AI-generated advice:_** There are risks if people or institutions become overly dependent on LLM outputs without sufficient human oversight, especially in high-stakes domains. The paper should explore safeguards and best practices.

4. **_Privacy and data protection:_** Using LLMs in healthcare and law raises major privacy concerns given the sensitive nature of the data involved. More robust discussion of privacy-preserving techniques is warranted.

5. **_Accountability and liability:_** It's unclear who would be held responsible for errors or harms caused by LLM outputs in these domains. The legal and ethical frameworks for AI accountability need more exploration.

6. **_Widening technology gaps:_** Smaller organizations or non-profits may lack access to advanced LLMs, potentially widening gaps in quality of services. Strategies for equitable access to these AI tools should be considered and discussed in the paper, in my opinion.

7. **_Impacts on human expertise:_** Over-reliance on LLMs could erode human domain expertise over time. Approaches for using LLMs to augment rather than replace human judgment should be emphasized.

8. **_Unintended consequences:_** Widespread LLM adoption could have unforeseen effects on financial markets, healthcare systems, or legal processes. More thorough analysis of potential systemic risks is needed.

9. **_Public trust and acceptance:_** Successful deployment requires building public trust in AI systems. The paper should discuss strategies for transparency and public engagement around LLMs in these domains.

**Claims And Evidence:**

Yes

**Requested Changes:**

I would highly recommend the authors to consider making the following changes to the manuscript:

1. Provide a clearer rationale for focusing specifically on finance, healthcare, and law. Explain why these domains were chosen over others like education or scientific research. (_critical for acceptance_)

2.  Add relatively short section (two to three paragraphs) clearly comparing the scope and contributions of this survey to previous related surveys.  (_critical for acceptance_)

3. Significantly reduce repetition and unnecessary full citations throughout. Only include full citations on first mention.  (_critical for acceptance_)

4. Add more critical analysis comparing different approaches rather than just listing recent work. Include more quantitative comparisons across methods where possible.  (_critical for acceptance_)

5. Expand the discussion of cross-domain insights and interactions between finance, healthcare, and law. Draw more connections and provide more critical analysis between challenges and opportunities across domains.  (_critical for acceptance_)

6. Provide a more in-depth and nuanced exploration of ethical considerations, potential risks, and unintended consequences of deploying LLMs in these high-stakes domains.  (_critical for acceptance_)

7. Add discussion of data quality issues, potential biases in training data, and approaches for mitigating these challenges in each domain.  (_critical for acceptance_)

8. Include more on the challenges of multilingual LLMs and applications in multilingual (non-English) contexts within these domains.  (_critical for acceptance_)

9. Address computational requirements and environmental impacts of training and deploying domain-specific LLMs.  (_critical for acceptance_)

10. Expand on the need for interpretable and explainable models in these domains where understanding decisions is often legally or ethically required.

11. Provide more details on how the promised reading list will be maintained and updated long-term.

12. Tighten the writing overall: Please remove redundant or repeated content, eliminate unnecessary capitalizations, rid of redundant and distracting citations, and, more importantly, make sure that all claims are properly supported.  (_critical for acceptance_)

13. Include a more rigorous and systematic methodology section describing how papers were selected for inclusion in the survey.

**Strengths And Weaknesses:**

1. **Comprehensive scope:** The paper provides a broad and thorough overview of LLM applications across three critical societal domains. This wide-ranging approach allows for valuable cross-domain comparisons and insights. By synthesizing research from these related  fields, the authors highlight common challenges and opportunities for cross-pollination of ideas. The comprehensive scope also makes this survey a valuable resource for researchers and practitioners looking to understand the landscape of LLM applications in these high-stakes domains.

2. **Structured organization:** The paper is well-organized with clear sections devoted to each domain, covering tasks, datasets, domain-specific LLMs, evaluations, and methodologies. This consistent structure across domains facilitates easy navigation and comparison. The logical flow from tasks to datasets to models to methodologies provides a coherent narrative of the research progression in each field. The use of subsections and bullet points further enhances readability and allows readers to quickly locate specific information of interest.

3. **Timely and relevant topic:** Given the rapid pace of LLM development and growing interest in domain-specific applications, this survey is highly relevant and timely. The authors have captured very recent developments, including recent models and papers from 2023 and early 2024. This **almost** up-to-date coverage is particularly valuable in a fast-moving field where new breakthroughs can quickly render older surveys obsolete. The timeliness of the survey positions it as a valuable resource for researchers looking to understand the current state-of-the-art and identify promising avenues for future work.

4. **Identification of key challenges:** The authors do a good job of highlighting key challenges in applying LLMs to these domains, such as data scarcity, need for domain expertise, and regulatory compliance. By framing these challenges clearly, the paper helps set the agenda for future research. (However, as I mention later on, these discussions sometimes feel too surface-level.)

5. **Helpful discussion of future directions:** The paper provides some high-level discussions of future research opportunities in each domain, but sometimes these forward-looking sections feel like listing potential topics rather than offering substantive analysis of why certain directions are promising and how they might address current limitations.

6. **Consideration of ethical implications:** The inclusion of a dedicated section on ethical concerns demonstrates awareness of the broader implications of deploying LLMs in high-stakes domains. This is particularly crucial given the potential for significant real-world impact in areas like healthcare diagnostics or legal decision-making. While the ethical analysis could certainly be more in-depth, its inclusion reflects an important recognition of the need to consider the societal consequences of AI advancements.

7. **Useful visualizations:** The figures summarizing tasks and datasets for each domain (e.g., Figure 1 for finance, Figure 3 for healthcare, and Figure 5 for law) provide helpful visual overviews. These figures nicely condense a large amount of information into easily digestible formats, allowing readers to quickly grasp the landscape of research in each domain. The consistent use of such figures across domains also facilitates cross-domain comparisons.

8. **Extensive coverage of datasets:** The paper provides a comprehensive overview of relevant datasets for each domain, including both widely-used benchmarks and more recent, specialized datasets. This thorough cataloging of datasets is valuable for researchers looking to quickly learn existing models or datasets in these domains.

9. **Balanced coverage of different modeling paradigms:** The paper covers a range of modeling approaches, from traditional fine-tuning to more recent developments like instruction tuning and retrieval-augmented generation. This balanced coverage provides a comprehensive view of the evolving methodologies in the field and helps readers understand the trade-offs between different approaches.

**Weaknesses:**

1. **Lack of critical analysis:** The paper often reads as a surface-level listing of recent work rather than providing deep, critical, and comprehensive analysis. There is limited comparative discussion of different approaches or critical evaluation of their strengths and limitations. For example, in the sections on domain-specific LLMs, the authors primarily describe various models without sufficiently analyzing their relative merits or shortcomings. A more rigorous comparative analysis would significantly strengthen the paper's contributions.

2. **Insufficient differentiation from prior surveys:** While related surveys are mentioned in Section 2, there is inadequate discussion of how this work differentiates itself or advances beyond previous surveys. The authors should more clearly articulate the novel contributions of this survey compared to existing literature reviews in each domain. A relatively short section (two to three paragraphs) dedicated to comparing the scope, methodology, and key insights of this survey versus previous ones would be helpful.

3. **Limited justification for domain selection:** The authors do not adequately explain why they chose to focus specifically on finance, healthcare, and law, rather than other important domains like education, scientific research, or journalism. A clearer rationale for this selection would strengthen the paper's framing.

4. **Lack of rigorous methodology:** The paper does not clearly describe the methodology for selecting papers for inclusion or how comprehensively the literature was surveyed. Without a systematic approach to paper selection and analysis, there is a risk of bias or incompleteness in the review. The authors should outline their search strategy, inclusion/exclusion criteria, and analytical framework.

5. **Overuse of citations and repetition:** The paper frequently includes full citations unnecessarily, even for concepts already introduced. This disrupts readability without adding value. Additionally, there is significant repetition of concepts and phrases throughout the paper. For example, "chain-of-thought (CoT)" is repeatedly introduced with a full citation. The writing could be tightened for clarity and conciseness.

6. **Limited quantitative comparisons:** While some performance comparisons are included, there is a lack of comprehensive quantitative analysis comparing different LLM approaches across tasks and domains. Including more tables or charts not only summarizing but also comparing the insights in these quantitative results from various studies would provide a clearer picture of what has been achieved and what has yet to be solved and addressed in each domain.

7. **Unsupported assertions:** Some claims are made without sufficient evidence or justification. For instance, statements about the potential impact of LLMs on various domains are often presented without backing from empirical studies or expert opinions. The authors should be more careful to support their claims with appropriate evidence.

8. **Inconsistent formatting and arbitrary capitalizations:** There are numerous formatting inconsistencies and unnecessary capitalizations throughout the paper (e.g., "X-ray Radiography, Ultrasound, Computed Tomography (CT) and Magnetic Resonance Imaging (MRI)"). These issues detract from the paper's professionalism and readability.

9. **Insufficient exploration of real-world deployment challenges:** While challenges are noted, there is inadequate discussion of the practical considerations for deploying LLMs in these high-stakes domains. More attention should be given to issues like regulatory compliance, integration with existing systems, and user acceptance.

10. **Questionable feasibility of promised reading list:** The authors' commitment to maintaining an updated reading list, while potentially valuable, raises questions of long-term feasibility that are not adequately addressed. The paper should provide more details on how this resource will be maintained and updated over time.

11. **Lack of critical perspective on ethical issues:** While the paper includes a section on ethical considerations, the discussion lacks depth and critical analysis. Given the high-stakes nature of these domains, a more nuanced exploration of potential risks, biases, and unintended consequences of LLM deployment is warranted.

12. **Insufficient discussion of cross-domain insights:** Despite covering three related domains, the paper misses opportunities to draw insightful comparisons or identify common themes and challenges across finance, healthcare, and law. A more integrated analysis could provide valuable insights for researchers working at the intersection of these fields.

13. **Lack of discussion on domain interactions:** The paper treats each domain largely in isolation, without exploring how advances in LLMs might impact the intersections between these fields (e.g., healthcare law, financial regulations in healthcare). Considering these interactions could provide a more holistic view of LLMs' societal impact.

14. **Lack of critical discussion on data quality and biases:** The paper does not sufficiently address the challenges of data quality and potential biases in training datasets for these domains. Given the high-stakes nature of finance, healthcare, and law, a more in-depth examination of these issues is warranted.

15. **Limited exploration of multi-lingual challenges:** The paper primarily focuses on English-language models and applications, with limited discussion of the challenges and opportunities for LLMs in multilingual or non-English contexts within these domains.

16. **Insufficient discussion of compute requirements and environmental impact:** Given the large-scale nature of LLMs, the paper should address the computational resources required for training and deploying these models in domain-specific contexts, as well as the associated environmental concerns.

17. **Lack of discussion on model interpretability and explainability:** While briefly mentioned, the paper does not adequately explore the crucial need for interpretable and explainable LLMs in these high-stakes domains, where understanding model decisions is often legally or ethically required.

18. **Limited exploration of potential negative societal impacts:** The paper could benefit from a more critical examination of potential negative consequences of widespread LLM adoption in these domains, such as job displacement, exacerbation of existing inequalities, or over-reliance on AI-generated advice.

---

> ### Author Response · Authors · 2024-09-06
> **Response to Reviewer Tyj3**
>
> Thank you for your thorough review and valuable feedback. We have carefully considered each comment and have made the following response and corresponding revisions:
>
> # Q: Why these domains were chosen
> A: We select the domains of finance, healthcare, and law (FHL) due to the following reasons:
> - **Significance of the Chosen Domains** The domains of FHL were selected because they are fundamental to societal welfare and function, each playing a pivotal role in everyday life and within larger economic and social frameworks. As we stated in Section 1, these domains are often grouped in academic and professional discussions due to their common features: they demand substantial professional knowledge, handle sensitive and confidential data, utilize complex multimodal documents, are subject to high legal stakes and rigorous regulatory oversight, and require a strong emphasis on explainability and fairness.
> - **Scope and Feasibility** It would be impractical and less impactful to encompass every possible domain where LLMs might be applied. Our objective is to provide a focused, in-depth analysis rather than a superficial overview that spans numerous fields. Choosing a limited number of domains sharing common features allows for a more thorough investigation of specific challenges and advancements, making the insights more valuable for both academic and practical implementations.
> - **State of Research and Need for a Survey** Our selected domains—FHL—stand out because they have a substantial body of existing research that has not yet been comprehensively synthesized in a high quality survey. This contrasts with fields like mathematics, which already have numerous surveys and reviews, diminishing the value of another summary. On the other hand, some emerging fields, e.g., neural science, materials science, etc., do not yet have enough mature research to support a full survey. The chosen domains are at an optimal point for a survey: there is enough existing work to provide a robust review, meanwhile, there are still many unsolved problems, concerns, and social implications for utilizing LLMs in real-world applications, which all merit a detailed discussion. This makes our survey both necessary and timely, as it can significantly influence ongoing research and application in these pivotal fields.
>
> # Q: Comparing this survey to previous related surveys
> A: In the section “Related Surveys”, we have listed other existing surveys related to our work and discussed our uniqueness. To highlight the differences between these surveys and ours, we added a paragraph “differences from existing surveys” for clarification, which include differences in scope, depth, and contribution.
> - **Scope** Unlike existing surveys that explore LLMs within NLP tasks or general applications, our study uniquely focuses on LLMs across three critical societal sectors FHL. Our study not only offers a unified high-level overview of the common ground of FHL, but also provides an in-depth review within each sector. This dual perspective ensures that our paper stands apart from general LLM surveys, as well as studies in single application domains.
> - **Depth** Our survey delves deeper than existing literature within each domain, including a thorough review covering tasks, techniques, evaluations, future prospects, and domain-specific ethics. The depth and breadth are both more extensive and integrative than existing related surveys.
> - **Contribution** (1) As far as we know, we are the first to provide a comprehensive view of LLM across the FHL sectors. (2) Our work meticulously reviews and organizes existing research into a well-structured categorization that spans problems, methods, experiments, and ethical discussions specific to each sector. (3) We outline promising future research directions in LLM studies within the FHL areas.
>
> # Q: Add more critical analysis comparing different approaches
> A: For the finance domain, our survey comprehensively evaluates popular methodologies across four key tasks: sentiment analysis, headline classification, named entity recognition, and question answering. Detailed quantitative comparisons of these methods are presented in Table 2 and Figure 2. We utilize five widely recognized datasets for these evaluations, which have been validated by the research community for their high quality and are frequently employed as benchmarks.
> For the healthcare domain, we provide quantitative comparisons and analysis for four tasks: Abnormality and Ambiguity Detection, Medical Report Generation, Medical Free-form Instruction Evaluation, and Medical-Imaging Classification Via Natural Language (table 4,5,6,7).
> For the law domain, we offer a comparison and analysis of three tasks: Legal Question Answering, Legal Event Detection, and Legal Judgment Prediction. Table 9 presents a detailed quantitative performance comparison of the mainstream LLMs. The  Legal Judgment Prediction task covers both fact-based articles and scene-based articles.

---

> > ### Author Response · Authors · 2024-09-06
> >
> > # Q:  Cross-domain insights
> > We recognize the value of exploring the potential synergies between these fields. However, our survey has found that the current state of research within these individual domains is still evolving. Although progress has been made, the maturity level of the LLM technologies in these sectors is not yet at a point where their integration into real-world applications is widespread or fully reliable. The current stage of development in each separate domain suggests that a deeper, more comprehensive exploration of cross-domain interactions may be premature at this stage. To the best of our knowledge, we have not identified any existing work on cross-domain interactions across these domains. Nonetheless, we acknowledge the potential of this direction for future research. It will become increasingly critical to understand how these fields can inform and enhance one another:
> >
> > What tasks and applications can benefit from such cross-domain interactions? For these tasks and applications, do they really need knowledge integration across different domains? And to what extent do we need such knowledge integration? These all depend on the specific applications.
> > For example, for healthcare law, it is critical to think about how to navigate the complexities of healthcare law, particularly concerning data privacy regulations such as HIPAA in the United States or GDPR in Europe. Training and deployment of LLMs should adhere to healthcare regulations and legal standards. For financial regulations in healthcare, the intersection of finance and healthcare is critical, particularly in areas like insurance, billing, and reimbursement. It is interesting to see how LLMs can assist in interpreting and applying complex financial regulations in healthcare, such as the Affordable Care Act or Medicare policies. For future prospects, LLMs may help assist financial decision-making processes within healthcare institutions, ensuring that financial practices align with both regulatory and ethical standards.
> >
> > # Q: More in-depth ethical considerations
> > We appreciate your suggestion. We added a more in-depth discussion in Section 6.2.
> >
> > At a high level, ethical considerations are essential for the whole AI community, and have been widely studied under different contexts. For general ethic problems such as safety, explanation and robustness (as mentioned in Section 6.1.2), FHL studies can certainly follow and extend existing lines of ethic studies in proper context. We show a clear overview of these general ethical considerations in Section 6, but it is not necessary to delve into general ethical studies, as they are not focused on FHL domains and do not represent the core objective of this wor, but we cited high-quality general ethic surveys to provide references for readers.
> >
> > As for LLM ethics in domain-specific applications (Section 6.1.3), first, the current studies in this area remain remarkably scarce and underexplored, with no systematic work. Consequently, our primary aim is to collect and organize existing works while looking forward to future directions. To achieve this, we have integrated our Section 6 with the following resources: (1) domain-specific ethics (highlighting the unique concerns in each domain, but may not be LLM-focused), and (2) a few existing explorations of LLM ethics within the FHL domains. Looking forward, we point out some urgent problems which have a broader impact in these areas, outline potential future directions for addressing these ethical challenges, and aim to foster a deeper and more responsible exploration of ethical considerations within these crucial sectors.
> >
> > # Q: Expand on the need for interpretable and explainable models in these domains
> > A: We have discussed the general considerations and methods for explainability in Section 6.1.2. In many settings, LLM applications in FHL can inherit or extend general LLM explanation methods. On top of that, in special FHL context that general explanation methods cannot tackle, some promising directions may include:
> > - **Domain-specific customization**: Tailor explanations to the specific needs and comprehension levels of different stakeholders in FHL sectors, recognizing that financial experts, healthcare providers, and legal professionals may require different types of information to effectively trust and use AI outputs.
> > - **Feedback loops for continuous improvement**: As FHL domains often involve long-term or multi-stage processes, we need to establish mechanisms for users to provide feedback on explanations, which can be used to continually refine explanation methodologies.

---

> > > ### Author Response · Authors · 2024-09-06
> > >
> > > # Q: Add discussion of data quality issues, potential biases in training data, and approaches
> > > A:
> > > For the finance domain, we made the discussion about data issue in Section 3.5. To summary, due to the difficulty of interdisciplinary collaboration and high cost of data annotation for specialized domains, we believe existing datasets generally suffer from low quality (annotation errors, low diversity, etc), and biases towards oversimplicity and standardization but not reflecting complex scenarios in real world, e.g., the FinQA dataset only covers simple calculations mostly one or two steps. To mitigate such issues, we suggest closer and more effective collaborations with financial experts, as well as acquiring comprehensive knowledge of the current status of the finance domain for AI and NLP researchers. A solid understanding of the knowledge of the both sides is necessary for us to tackle real challenges and develop realistic datasets. In the era of LLMs, the requirement for large-scale training data is no longer a must, somewhat reducing the costs associated with data annotation. However, the expenses and logistical efforts involved in curating high-quality test data and maintaining productive collaborations with domain experts remain significant hurdles in creating realistic, high-quality datasets, which requires enough research funding, sufficient time commitment and effective management.
> > >
> > > For the healthcare domain. Data Quality: Healthcare data often suffer from inconsistencies, missing information, and variability in data formats, which can significantly impact the performance of LLMs. For example, electronic health records (EHRs) may contain incomplete patient histories or inconsistent coding practices across different providers. Potential Biases: Biases in healthcare data can arise from historical inequalities, such as underrepresentation of certain demographic groups in clinical trials or disparities in healthcare access. These biases can lead LLMs to make less accurate predictions or recommendations for underrepresented groups, exacerbating health disparities. Mitigation Approaches: We will discuss methods for improving data quality, such as standardizing data entry practices and using data augmentation techniques to address missing information. Additionally, we will explore bias mitigation strategies, including re-weighting training data, employing fairness-aware algorithms, and conducting regular bias audits to ensure that LLM outputs do not disproportionately affect certain groups.
> > >
> > > For the law domain, we discussed data issues in Section 5.5. Data Quality: Most of the existing legal datasets collected in the natural environment are incomplete, sparse, and complex. Their complexity and academic nature make it difficult for conventional deep learning methods to provide annotations, while manual annotation in the legal domain requires higher standards and costs compared to general fields, which significantly affects the performance of LLMs. For example, legal documents, judgments, and statutes from different jurisdictions may use different legal terminologies or formats, and even within the same case, legal interpretations may vary due to differences in expression between lawyers and judges. Potential Bias: Bias in legal data may stem from historical inequalities in legal rulings, law enforcement, and judicial processes. For example, the representation of different social groups in the criminal justice system may be biased, and certain vulnerable groups, such as individuals who lack legal expertise or economic resources to seek human legal assistance, may face unjust treatment, resulting in these biases being embedded in LLM training. Mitigation Methods: To address these issues, we will explore ways to improve the quality of legal data, such as promoting the standardization of legal terminology and document formats, and using data cleaning and augmentation techniques to fill in missing information or resolve inconsistencies. For instance, CUAD, a standardized dataset created by dozens of legal experts from the Atticus Project, contains over 13,000 annotations. Additionally, we will discuss bias mitigation strategies to ensure that LLMs produce fairer and legally compliant predictions for different social groups.

---

> ### Author Response · Authors · 2024-09-06
>
> # Q: How the reading list will be maintained and updated long-term
> A: To ensure the long-term maintenance and contribution of our reading list, we will:
> - **Keep updating new papers**: We will continue to update the reading list to keep it up-to-date.
> - **Open access**: The reading list will be fully open in public. The researchers are highly encouraged to read, check, and provide feedback for it.
> - **Other contributors**: We encourage other contributors to join and add to the reading list.
> - **Quality check and bias mitigation**: We develop rigorous quality checks on the selected papers for our reading list to ensure high standards and eliminate any potential bias.
>
> # Q: Describe how papers were selected for inclusion in the survey
> A: To ensure the quality of our survey, we employed a rigorous and systematic methodology for selecting papers:
> - Relevance: Papers should be highly related to the survey’s scope of LLM + FHL areas.
> - Quality: Papers should be published in peer-reviewed venues known for high standards, or preprints (e.g., arxiv) with quality evaluation by our authors.
> - Recency: Priority was given to more recent publications to ensure the survey’s timeliness, though seminal works were also included.
> - Impact: Papers with significant citations or those that have clearly influenced the field were prioritized.
>
> # Q: Broader impact concern
> For broader impact concerns, we added further discussion "Further Ethical Concerns in FHL" in Section 6.2 to include them as further ethical problems. For each concern, we introduced our vision for future potential solutions.
>
> # Q: Writing problems
> Thanks for pointing that out. We have checked our paper to correct writing problems.

---

### Review · Reviewer_LqKo · 2024-08-23

**Summary Of Contributions:**

This survey paper summarizes the state of domain-specific LLMs in finance, medicine, and law, draws shared connections across these settings for ethnical condiderations and desirdata for real-world use (e.g. "explainability"). The paper summarizes existing domain-specific LLMs, their pretraining datasets, tasks, and evaluation techniques.

This work contributes to a growing body of surveys covering LLMs in domain-specific settings. I focus most of my comments on the medical LLM section.

**Audience:**

Yes

**Broader Impact Concerns:**

No concerns.

**Claims And Evidence:**

Yes

**Requested Changes:**

- Comment on space of LLMs trained and evaluated on electronic health record (EHR) data. This is mentioned in passing in section "Future Prospects" but there are prior survey papers for LLMs that explore this topic in more depth and the gaps therein.

  - Wornow et al 2023. "[The shaky foundations of large language models and foundation models for electronic health records](https://www.nature.com/articles/s41746-023-00879-8)"

- Include more surveys outside the purely technical realm to better characterize how the medical domain is framing and thinking about the challenges of deploying generative AI more broadly.

- Include specific citations outlined in the weaknesses above.

**Strengths And Weaknesses:**

**Strengths**

- Covers the wide range of recent medical LLMs
- Captures general NLP categories across the 3 discussed LLM domains
- Outlines some ethnical connections across domains to expand the survey


**Weaknesses**

- The manuscript covers a lof of material, 3+ deep topics on finance, medicine, law use cases of LLMs and corresponding ethical implications. The manuscript almost feels like 2 papers combined: a survey of current work on domain-specific LLMs and a paper on the shared ethical principles (Figure 6) of high-stakes AI. I do feel this paper tries to cover too much material at the cost of focusing on clearer story for the reader.

- As the authors note, there are many such recent LLM surveys covering these areas. The manuscript makes arguments for the unique contributions and need for this work.

> "Our work encompasses a broader spectrum, surveying various applications in both the pure NLP domain and multimodal scenarios. We also discuss recent novel tasks such as medical instruction following and medical imaging classification via natural language."

However, I feel there have been a number of recent surveys, especially in the medical and digital medicine literature, that provide a broader view than just the classic NLP (infoextract, sentence classification, QA) and vision-language model spaces in medicine as covered here. The survey would be enriched by bringing in more domain-focused discussions of LLM use in medicine. Some example recent survey papers and perspectives worth including.

 - Bedi et al. 2024. [A Systematic Review of Testing and Evaluation of Healthcare Applications of Large Language Models (LLMs)](https://www.medrxiv.org/content/10.1101/2024.04.15.24305869v4) Preprint.

- Omiye  et al. 2024. [Large Language Models in Medicine: The Potentials and Pitfalls: A Narrative Review](https://www.acpjournals.org/doi/full/10.7326/M23-2772?casa_token=EyotCeeolAYAAAAA:qdUVM6EJ-V1kD50n87vzUAZk2a15R0ZHT8WPbzCanz9E3j8Wl-TvoC83zn0cZj7AtpeF2dYpeUlR) Annals of Internal Medcine.

- Peng et al 2023. [A study of generative large language model for medical research and healthcare](https://www.nature.com/articles/s41746-023-00958-w). npj Digital Medicine

- Shah et al 2023. [Creation and Adoption of Large Language Models in Medicine](https://jamanetwork.com/journals/jama/article-abstract/2808296). JAMA.

> 4.4 Medical Report Generation

- There is work demonstrating how lexical measures of text generation (e.g., BLEU, ROUGE-L, METEOR) and info-extract-style evaluations of generated text (where some set of concepts are labeled in generated text) don't correlate well with expert judgement of generation quality (Xie et al 2024). Given this, I'm curious as the motivation and take-aways for Table 5.

 - Xie et al 2024. [DocLens: Multi-aspect Fine-grained Evaluation for Medical Text Generation](https://aclanthology.org/2024.acl-long.39/) ACL 2024.

> 4.5 Medical Free-form Instruction Evaluation

- The MedAlign dataset (Fleming et al 2024), 303 instructions written by clinicians for tasks defined over longitudinal EHRs, is also worth citing as a medical instruction benchmark targeting real-world medical applications.

- Fleming et al 2024. [MedAlign: A Clinician-Generated Dataset for Instruction Following with Electronic Medical Records](https://ojs.aaai.org/index.php/AAAI/article/view/30205). AAAI 2024.


> "As a novel task proposed recently, existing LMs may not readily include Under review as submission to TMLR
 such a task into its pre-training stage. Therefore, evaluation of this task allows us to investigate how language models perform for unseen tasks."

- I'm not really sold on the merits of the Table 4 experiment results, showing how fine-tuned smaller BERT style models work better for sentence classification (abnormality / ambiguity classes) vs. zero and few shot autoregressive LLMs. Ignore missing experimental details (how many shots?) the take-away "there is still a gap between finetuned LMs and prompted LLMs" is well-established in the literature.

> "The evaluation of LLMs, such as ChatGPT, on medical exams, including US Medical Exams (Kung et al., 2023) and Otolaryngology–Head and Neck Surgery Certification Examinations (Long et al., 2023), indicates that they achieve scores close to or at the passing threshold. This suggests the potential of LLMs to support real-world medical usages such as medical education and clinical decision-making"

- There is a strong argument that passing medical exams is viewed as a very weak, at least one that lacks ecological validity. While curated medical vignettes and other exam-style assessments of LLMs provide some insight into knowledge skills (either from weight memory or via retrieval abilities) it is far removed from the majority of tasks involved with the practice of medicine.

---

> ### Author Response · Authors · 2024-09-06
> **Response to Reviewer LqKo**
>
> ## Q: paper covering too much material
> A: Thank you for comment, we understand the concern. However, we would like to clarify our intent and the cohesive structure of the paper:
> Combination of LLM+FHL and ethics: The intrinsic connection between the technical aspects of LLM and ethical considerations in the finance, health, and law (FHL) sectors necessitates their combined discussion. Ethics is not merely an adjunct topic but a fundamental component that must be addressed in tandem with technological advancements.
> Distinct focus on domain-related ethics: Unlike general ethical surveys, the ethical discussion in our manuscript deliberately narrows its focus to domain-related concerns. This target allows us to delve deeper into the unique ethical dilemmas and considerations directly within FHL sectors, and we do not overextend the scope but rather intensify the discourse within a clearly defined boundary.
>
> ## Q: Include more citations in medicine-related LLM.
> A: Thanks for your suggestion, we will include the citations as mentioned in the review and discuss them in the revision.
>
> ## Q: Metrics and Take-aways for Table-5
> A: The evaluation of text generation has long been an active research topic. Though n-gram metrics are not the best metrics in terms of high agreements with medical experts, they still have a level of correlation to represent the quality of generated text. We include them as they are commonly reported in previous papers. We also report clinical efficacy to measure the clinical accuracy of the generated reports.
>
> ## Q: Table 4 experiment results
> A: The few-shot results are evaluated with 5-shots. We will clarify in the paper. As for the overall interpretation of the results, one argument is “prompted LLMs perform worse on novel tasks (tasks that are most likely unseen during pre-training)”.
>
> ## Q: While curated medical vignettes and other exam-style assessments of LLMs provide some insight into knowledge skills, it is far removed from the majority of tasks involved with the practice of medicine.
> A: We acknowledge this argument. Passing medical exams does not mean LLMs are ready to be deployed for medical education or clinical decision-making. We will add this comment into the paper. On the other hand, though these medical exams are not fully capturing real-world practice, as of today, they may still be useful for assessing foundational medical knowledge and reasoning skills in LLMs.
>
> ## Q: Comment on the space of LLMs trained and evaluated on EHR data.
> A:  Thank you for your valuable suggestion. We will include a dedicated subsection on LLMs for EHR in the revised version and reference the suggested survey paper. Below is a brief summary of current work on LLMs in this area:
> Several LLMs in healthcare have been further fine-tuned on electronic health record (EHR) data to improve clinical outcomes, such as predicting 30-day readmission, mortality, and extended lengths of stay on clinical notes, or multimodal data such as time series and text [1]. For instance, Gema et al. [2] performed parameter-efficient fine-tuning of the LLama model on clinical notes from MIMIC-IV to enhance clinical knowledge injection. Additionally, Fleming et al. [3] introduced MedAlign, an EHR-based instructional dataset, to promote the development of LLMs for healthcare applications. Despite these advancements, the use of LLMs on EHR data remains limited due to high costs and constrained data availability [4]. We will provide a detailed description of the work in this subdomain in the revised manuscript.
> [1] Zhang et al. (2023). Improving Medical Predictions by Irregular Multimodal Electronic Health Records Modeling.
> [2] Gema et al. (2023). Parameter-Efficient Fine-Tuning LLaMA for the Clinical Domain.
> [3] Fleming et al. (2024). MedAlign: A Clinician-Generated Dataset for Instruction Following with Electronic Medical Records.
> [4] Johnson et al. (2023). MIMIC-IV, a Freely Accessible Electronic Health Record Dataset.
>
>
> ## Q: Include more surveys outside the purely technical realm about how the medical domain is thinking about and deploying generative AI.
> A: Thanks for your suggestion! We will cite more surveys outside the purely technical realm. If you have any specific papers in mind, feel free to let us know.

---

### Comment · Action_Editor_ioe2 · 2024-08-29
**Author discussion period halfway mark**

Dear Paper2866 Authors,

With the release of all reviews, please note that we are approaching the halfway mark for the author discussion period (more details [here](https://jmlr.org/tmlr/author-guide.html)).  Please feel free to reply to/clarify/discuss reviewer concerns and remarks throughout this period, after which a decision will be made about the submission.

Best,

Paper2866 Action Editor

---

### Decision · Action_Editor_ioe2 · 2024-10-03

**Recommendation:** Accept with minor revision

**Comment:**

All the reviewers agree that the manuscript exhaustively covers existing LLM work in Finance, Healthcare, and Law (FHL).  The majority of reviewers vote in favor of acceptance.  The major merit of this survey is its comprehensiveness, which could be a valuable onboarding route for any TMLR readers interested in beginning research in either FHL field.

There were concerns of the writing at times, with Reviewer Tyj3 suggesting the writing could be tighter (which I agree with).  I currently recommend accepting with minor revisions, where I am requesting the authors please perform an editorial pass to polish the existing manuscript.  For example, once an acronym is defined, please use the acronym henceforth (e.g., see Chain-of-Thought (CoT) in the paper).  Also, please make sure terms are standardized (CoT appears as Chain-of-Thought, Chain of thought, and chain of thought; please use only one).

**Audience:**

Yes

**Claims And Evidence:**

Yes

---

> ### Comment · Action_Editor_ioe2 · 2024-11-05
> **Requested minor revisions**
>
> Dear authors,
>
> Could the authors please go through the manuscript and perform the minor revisions requested upon acceptance?
>
> Thank you,
>
> AE for Submission 2866

---

> > ### Author Response · Authors · 2024-11-08
> > **Updated submission**
> >
> > Dear AE and reviewers,
> >
> > Thanks for your kind reminder. We have made modifications to the latest submission version to address the concern, including the items we mentioned in our responses. The major changes can be summarized as follows:
> >
> > - Including more comprehensive references of related work
> > - Clarifying the difference between our work and other surveys and highlighting our unique contributions
> > - Adding more critical content for methods and analysis
> > - Expanding discussions for ethical concerns and potential solutions
> > - Tightening the writing to remove repeated and redundant content.
> >
> > Again, we appreciate all the helpful comments you offered throughout the reviewing process.

---

> > > ### Comment · Action_Editor_ioe2 · 2024-11-08
> > > **Confirmation**
> > >
> > > Dear Authors,
> > >
> > > Thanks for the manuscript updates, approving the camera ready version.
> > >
> > > Best regards,
> > >
> > > AE